# Impact of the clinically approved BTK inhibitors on the conformation of full-length BTK and analysis of the development of BTK resistance mutations in chronic lymphocytic leukemia

Raji E Joseph[1]*[†], Thomas E Wales[2][†], Sandrine Jayne[3], Robert G Britton[3], D Bruce Fulton[1], John R Engen[2], Martin JS Dyer[3], Amy H Andreotti[1]*

[1]Roy J. Carver Department of Biochemistry, Biophysics and Molecular Biology, Iowa State University, Ames, United States; [2]Department of Chemistry and Chemical Biology, Northeastern University, Boston, United States; [3]The Ernest and Helen Scott Haematological Research Institute, Leicester Cancer Research Centre, College of Life Sciences, University of Leicester, Leicester, United Kingdom

*For correspondence:
jraji@iastate.edu (REJ);
amyand@iastate.edu (AHA)

[†]These authors contributed
equally to this work

Competing interest: The authors
declare that no competing
interests exist.

Reviewing Editor: Volker
Dötsch, Goethe University
Frankfurt, Germany

## eLife Assessment

The manuscript reports on an **important** comparison of a range of approved clinical inhibitors for BTK used for the treatment of chronic lymphocytic leukemia (CLL). The authors provide **compelling** evidence for their claims, using a combination of HDX-MS and NMR spectroscopy. The novelty is that this study also seeks to evaluate resistance mutation bias. The manuscript will be of high interest to scientists working on protein drug interactions.

**Abstract** Inhibition of Bruton's tyrosine kinase (BTK) has proven to be highly effective in the treatment of B-cell malignancies such as chronic lymphocytic leukemia (CLL), autoimmune disorders, and multiple sclerosis. Since the approval of the first BTK inhibitor (BTKi), Ibrutinib, several other inhibitors including Acalabrutinib, Zanubrutinib, Tirabrutinib, and Pirtobrutinib have been clinically approved. All are covalent active site inhibitors, with the exception of the reversible active site inhibitor Pirtobrutinib. The large number of available inhibitors for the BTK target creates challenges in choosing the most appropriate BTKi for treatment. Side-by-side comparisons in CLL have shown that different inhibitors may differ in their treatment efficacy. Moreover, the nature of the resistance mutations that arise in patients appears to depend on the specific BTKi administered. We have previously shown that Ibrutinib binding to the kinase active site causes unanticipated long-range effects on the global conformation of BTK (Joseph et al., 2020). Here, we show that binding of each of the five approved BTKi to the kinase active site brings about distinct allosteric changes that alter the conformational equilibrium of full-length BTK. Additionally, we provide an explanation for the resistance mutation bias observed in CLL patients treated with different BTKi and characterize the mechanism of action of two common resistance mutations: BTK T474I and L528W.

## Introduction

Ibrutinib (IMBRUVICA), has revolutionized the treatment of the B-cell malignancy, CLL (*Burger et al., 2015*). Ibrutinib is a covalent active site inhibitor of the multi-domain B-cell kinase, BTK (*Figure 1a*), that was first approved by the FDA in 2013 (*Davids and Brown, 2014*). Inhibition of BTK disrupts signaling downstream of the B-cell receptor (BCR), a pathway on which the survival of CLL cells is dependent (*Honigberg et al., 2010*; *Herman et al., 2011*). The success of Ibrutinib has spurred the development of other BTKi including the now clinically approved BTK inhibitors: Acalabrutinib (CALQUENCE), Zanubrutinib (BRUKINSA), Tirabrutinib/ONO-4059 (VELEXBRU), Pirtobrutinib/LOXO-405 (JAYPIRCA) and Orelabrutinib (HIBRUKA) (*Rozkiewicz et al., 2023*; *Shirley, 2022*; *Montoya and Thompson, 2023*; *Kueffer et al., 2021*; *Frustaci et al., 2023*). All of the clinically approved BTKi (with the exception of Pirtobrutinib) are covalent active site inhibitors that bind to BTK C481 residue within the kinase active site. Pirtobrutinib is currently the first and only clinically approved non-covalent BTK active site inhibitor (*Montoya and Thompson, 2023*). Importantly, all clinically approved BTK inhibitors in this study, covalent and non-covalent, completely block nucleotide binding.

While BTKi are highly effective in the treatment of CLL, they are also being used to treat other B-cell malignancies such as Mantle cell lymphoma (MCL), Waldenström's macroglobulinemia, Marginal zone lymphoma (MZL) and are being evaluated in the treatment of multiple sclerosis and rheumatoid arthritis (*Rozkiewicz et al., 2023*; *Mendes-Bastos et al., 2022*; *Robak and Robak, 2022*). The plethora of clinically approved BTKi along with several other promising candidates currently in clinical trials pose a new challenge with respect to patient treatment. We now need a way to assess the suitability of a given BTKi for a given patient or disease state. Indeed, recent clinical trial data comparing one BTKi to another indicate that BTKi differ in their effectiveness in treating various conditions (*Byrd et al., 2021*; *Brown et al., 2023*; *Lovell et al., 2022*; *Buske et al., 2023*). While toxicity, specificity, and other criteria are used in these clinical comparisons, a molecular-level characterization of the interaction of the inhibitor with full-length BTK is lacking and could be key to understanding these differences. In our previous work, using high-resolution structural biology techniques such as Nuclear Magnetic Resonance (NMR) and Hydrogen Deuterium Exchange Mass Spectrometry (HDX-MS), we have shown that some inhibitors (Ibrutinib) exert unanticipated allosteric effects upon binding to BTK while others (Fenebrutinib) do not (*Joseph et al., 2020*). Ibrutinib binding to the BTK kinase active site leads to a shift in the conformational ensemble of full-length BTK towards an open conformation where the regulatory SRC homology (SH3 and SH2) domains are released from the 'back' surface of the kinase domain, disrupting the closed autoinhibited conformation of the full-length protein (*Figure 1b*, *Joseph et al., 2020*). Allosteric conformational changes upon drug binding have been shown to alter protein-ligand interactions in other systems with functional consequences in vivo (*Sonti et al., 2018*; *Tong et al., 2017*; *Skora et al., 2013*). The full impact of the panel of currently approved BTKi on the conformation of full-length BTK is not known.

Despite the success of BTKi in CLL, a recurring problem is the development of BTKi resistance. Resistance typically develops in patients ~2 y after the start of BTK inhibitor treatment (*Sedlarikova et al., 2020*). Analysis of CLL patients that stop responding to BTKi has revealed that most develop mutations within *Btk* or in the substrate of BTK: Phospholipase C gamma 2 (PLCγ2) (*Maddocks et al., 2015*; *Ahn et al., 2017*; *Woyach et al., 2014*). Interestingly, the specific resistance mutations that develop in *Btk* seem to be dependent on the specific BTKi used (*Woyach et al., 2014*; *Blombery et al., 2022*; *Furman et al., 2014*; *Wang et al., 2022*; *Jackson et al., 2023*; *Handunnetti et al., 2019*; *Woyach et al., 2019*). While data available so far for the BTKi inhibitors Acalabrutinib, Zanubrutinib, Tirabrutinib, and Pirtobrutinib are low in number, there is an emerging trend (*Figure 1c and d*). Perhaps not surprisingly, patients treated with the reversible inhibitor Pirtobrutinib do not develop mutations in C481 (*Figure 1d*). Instead, they develop other kinase active site mutations including BTK T474I and L528W (*Figure 1c and d*; *Wang et al., 2022*). In contrast, 90% or more of CLL patients treated with Ibrutinib and Acalabrutinib develop mutations in BTK C481 (*Figure 1d*, *Supplementary file 1*, *Maddocks et al., 2015*; *Ahn et al., 2017*; *Woyach et al., 2014*). Substitution of C481 with serine is the most common resistance mutation found in these patients (*Woyach et al., 2017*). While mutation of BTK C481 to residues other than serine have also been reported (C481F/Y/R), they occur at a much lower frequency (*Nakhoda et al., 2023*). Additionally, other sites within BTK such as T474 and L528 are rarely (if ever) mutated in Ibrutinib and Acalabrutinib-treated CLL patients (*Figure 1d*, *Supplementary file 1*, *Maddocks et al., 2015*; *Woyach et al., 2014*; *Blombery et al., 2022*; *Woyach*

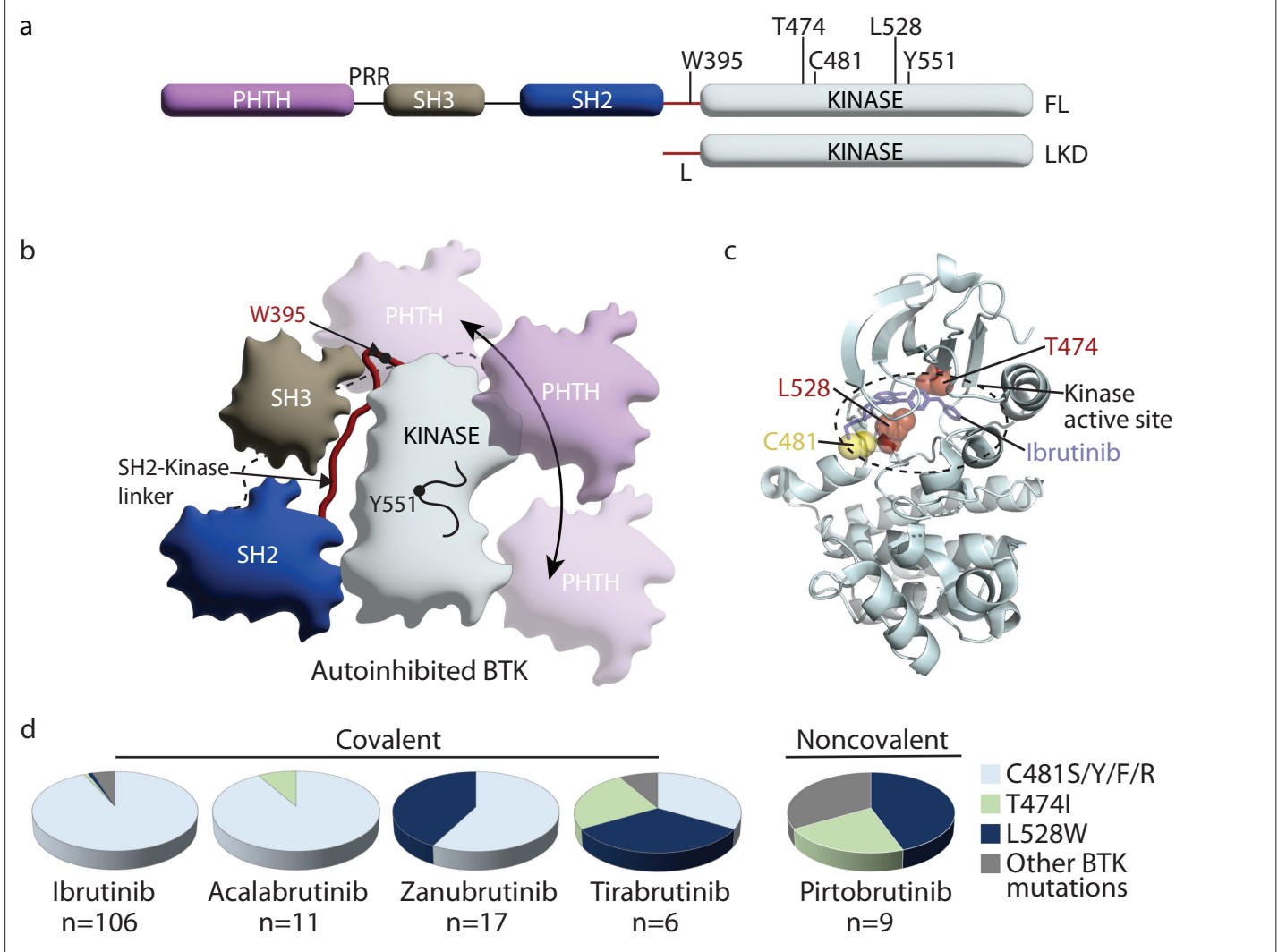

**Figure 1.** Inhibition of Bruton's tyrosine kinase (BTK) is an effective way to treat chronic lymphocytic leukemia (CLL). (**a**) Domain organization of full-length (FL) BTK and the BTK linker-kinase domain (LKD) fragment used in this study: PHTH, Pleckstrin homology-Tec homology domain; PRR, proline-rich region; SH3, Src homology 3 domain; SH2, Src homology 2 domain, SH2-kinase linker (L) and the catalytic kinase domain. Key residues are indicated above each domain. (**b**) Autoinhibited conformation of FL BTK based on the crystal structure of FL BTK (*Lin et al., 2024*). The PHTH domain (purple) is dynamic, in transient contact with several regions on the core SH3-SH2-kinase domain, and is not visible in the crystal structure of full-length BTK (*Lin et al., 2024*). Dynamics of the PHTH domain is represented by the multiple poses of the PHTH domain and the double-headed arrow. (**c**) Co-crystal structure of BTK LKD (light cyan cartoon) bound to Ibrutinib (PDB ID: 5P9J) showing the location of C481 (yellow spheres), T474, and L528 (red spheres) within the kinase active site (broken oval). (**c**) Pie charts showing the prevalence of the BTK resistance mutations in CLL patients treated with various BTK inhibitors. The total number of patients with mutations in BTK are indicated below each chart. See *Supplementary file 1* for additional details.

*et al., 2017*; *Kanagal-Shamanna et al., 2019*; *Sharma et al., 2016*). Surprisingly, BTK T474 and L528 are frequently mutated in CLL patients treated with the covalent inhibitors Zanubrutinib and Tirabrutinib (*Figure 1d*, *Blombery et al., 2022*; *Jackson et al., 2023*; *Handunnetti et al., 2019*). BTK T474I and L528W are found at almost equal or higher frequency compared to C481S in these patients (*Figure 1d*, *Supplementary file 1*, *Blombery et al., 2022*; *Jackson et al., 2023*; *Handunnetti et al., 2019*). Moreover, in Zanubrutinib-treated patients, the L528W mutation was often present together with the C481S mutation (on different alleles), and with the L528W mutation present at a higher allelic frequency compared to C481S (*Handunnetti et al., 2019*). This mutational bias with respect to covalent BTKi is unexpected given the shared mode of action across the panel of covalent inhibitors, and the reasons for these differences are unclear. Additionally, the mechanism of action of the BTK T474I and L528W mutations is not known. Given the development of T474I and L528W mutations in multiple CLL patients treated with both covalent (Zanubrutinib and Tirabrutinib) and non-covalent

(Pirtobrutinib) BTK inhibitors, we focused on these BTK mutations and explore their mechanisms of action.

Here, we probe the impact of four clinically-approved BTKi: Acalabrutinib, Zanubrutinib, Tirabrutinib, and Pirtobrutinib on the conformation of full-length BTK. We find that each of these BTKi brings about a unique combination of changes in full-length BTK. Acalabrutinib, Zanubrutinib, and Tirabrutinib disrupted the autoinhibited conformation of full-length BTK similarly to Ibrutinib but did so to varying degrees. Interestingly, Acalabrutinib and Tirabrutinib altered the dynamics of the kinase G-helix, a region that has been previously characterized as the PLCγ substrate docking site (*Xie et al., 2013*). Pirtobrutinib on the other hand stabilized the compact autoinhibited conformation of full-length BTK and is the first BTK inhibitor observed to do so. Additionally, we probed the mechanism of action of the BTK resistance mutations T474I and L528W. We show that the catalytically inactive BTK L528W mutant activated the SRC kinase HCK and that this activation is dependent on the proline-rich region within BTK. The BTK T474I mutation disrupted binding to Zanubrutinib, Tirabrutinib, and Pirtobrutinib and likely evades the action of these drugs due to reduced binding to these inhibitors. Furthermore, we provide an explanation for the mutational bias observed in patients treated with different covalent BTKi. The development of the C481S resistance mutation is dependent on the half-life of the inhibitor which likely explains the low prevalence of the C481S resistance mutation in patients treated with the covalent inhibitors Tirabrutinib and Zanubrutinib (which have a long half-life) as compared to Ibrutinib and Acalabrutinib (which have a short half-life). Characterization of the interaction of BTKi with full-length BTK allows us to better interpret clinical trial results and will help guide the choice of BTKi to be used for treatment. Furthermore, understanding the mechanism of action of resistance mutations allows us to develop treatment strategies that either prevent or delay the development of resistance mutations and ways to treat them when they arise.

## Results

### Assessing the impact of inhibitor binding on the isolated BTK linker-kinase domain by NMR

To evaluate the impact of inhibitor binding on BTK, we first monitored its effect on the conformation of the linker-kinase domain (LKD) fragment of BTK (*Figure 1a*). The catalytic kinase domain of BTK can adopt an active or inactive conformation in solution and previous studies have shown that inhibitor binding can stabilize one or more of these conformations (*Joseph et al., 2020*). The switch from an inactive to an active kinase conformation involves changes in key structural elements within the kinase domain. These changes include the inward movement of the C-helix from an inactive 'αC out' to the active 'αC in' position, changes in the side chain rotamer conformation of BTK W395, formation of a conserved BTK K430:E445 salt bridge and unfurling of the collapsed activation loop, leading to the exposure of the conserved Y551 on the activation loop for phosphorylation (*Figure 2a*, *Joseph et al., 2020*).

Crystal structures of the BTK kinase domain in complex with Acalabrutinib, Zanubrutinib, Tirabrutinib, or Pirtobrutinib are available and show that each of these BTK inhibitors stabilizes the kinase domain in an inactive conformation similar to that of Ibrutinib-bound BTK (*Figure 2b–f*, *Joseph et al., 2020*; *Lin and Andreotti, 2023*; *Gomez et al., 2023*; *Guo et al., 2019*). In all the inhibitor-bound structures the C-helix is in the 'αC out' position and the activation loop is collapsed into the kinase active site, burying the conserved BTK Y551 (*Figure 2b–f*). Moreover, the superposition of these BTK/inhibitor complexes shows that the structures of the kinase domain remain largely the same regardless of which inhibitor is bound (*Figure 2g*). The structure of the Tirabrutinib bound BTK is the only structure that shows a difference; the activation loop in that complex adopts an alternative inactive loop conformation compared to the other inhibitor bound structures (*Figure 2e and g*).

Previous NMR and HDX-MS analysis of the interaction of BTK with a different panel of active site inhibitors has shown that there are differences between the solution and crystal behavior (*Joseph et al., 2020*). To test whether the BTK inhibitors Acalabrutinib, Zanubrutinib, Tirabrutinib, and Pirtobrutinib can stabilize an inactive BTK kinase conformation in solution as predicted by the crystal structures, we evaluated the BTK inhibitor bound complexes by NMR. The LKD fragment of BTK was isotopically labeled with $^{15}N$ and a $^1H$-$^{15}N$ TROSY-HSQC spectrum was obtained in the presence or absence of the inhibitor (*Figure 3*). We have previously shown that BTK W395 within the linker region

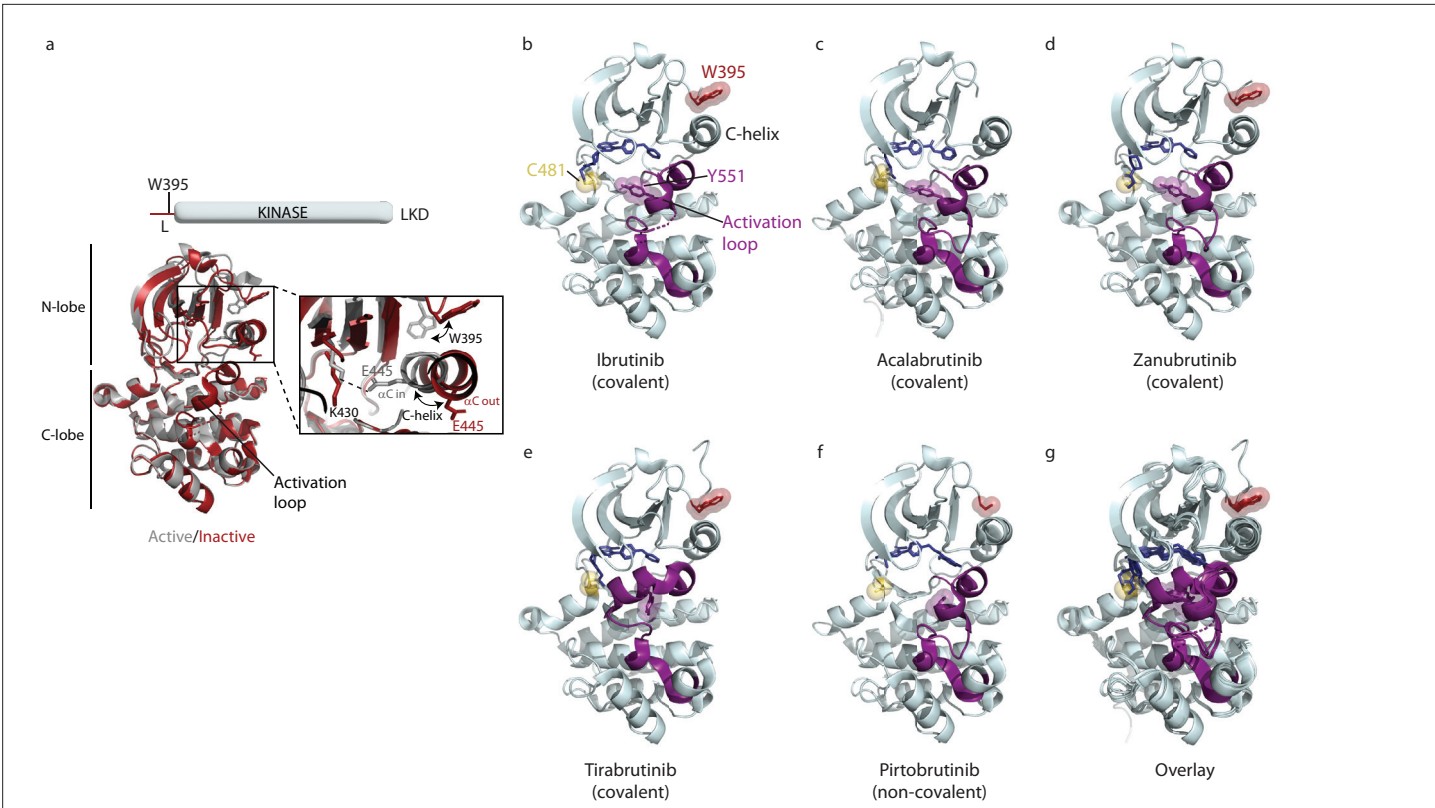

**Figure 2.** The Bruton's tyrosine kinase (BTK) kinase domain can interconvert between active and inactive conformations. (**a**) Superposition of the structure of BTK linker-kinase domain (LKD) bound to Dasatinib (PDB ID: 3K54) in the active kinase conformation (gray cartoon) with the Ibrutinib bound structure (PDB ID: 5P9J) in the inactive conformation (red cartoon). The expanded inset shows the inward movement of the αC-helix, the change in W395 rotamer conformation, and the K430/E445 salt bridge formation that accompanies kinase activation. (**b–f**) Co-crystal structures of BTK LKD (light cyan cartoon) bound to Ibrutinib (PDB ID: 5P9J), Acalabrutinib (PDB ID: 8FD9), Zanubrutinib (PDB ID: 6J6M), Tirabrutinib (PDB ID: 5P9M) and Pirtobrutinib (PDB ID: 8FLL) in the inactive kinase conformation. The inhibitors are shown as dark blue sticks, the kinase activation loop is purple and C481, Y551, and W395 residues are shown as sticks with transparent spheres. Electron density for part of the activation loop is missing in the Ibrutinib co-crystal structure and is indicated as dotted lines (**b**). In the Acalabrutinib structure, the activation loop has several mutations (*Lin and Andreotti, 2023*) and the SH2-kinase linker (including W395) is absent (**c**). Electron density for the W395 sidechain is missing in the BTK:Pirtobrutinib co-crystal structure (**f**). (**g**) Overlay of the BTK:Ibrutinib, Acalabrutinib, Zanubrutinib, Tirabrutinib, and Pirtobrutinib co-crystal structures. With the exception of the Tirabrutinib co-crystal structure (**e**), no major structural variation is observed in the kinase domains. The activation loop in the Tirabrutinib bound structure adopts a different conformation compared to the other co-crystal structures.

(L) that precedes the kinase domain provides a useful probe to monitor the conformational state adopted by the kinase domain in solution (*Joseph et al., 2020*; *Joseph et al., 2017*). Assignments for the apo BTK linker-kinase domain, which adopts the active (αC in) conformation, show that the W395 indole $^1$H resonates at 10.21 ppm in the $^1$H-$^{15}$N TROSY-HSQC spectrum (*Figure 3*, top panel, black spectrum). An upfield shift in the BTK W395 side chain indole NH resonance is observed upon Ibrutinib binding and is consistent with the outward movement of the C-helix (αC out), and stabilization of the inactive kinase domain conformation by Ibrutinib (*Figure 3*, top panel, cyan spectrum *Joseph et al., 2020*).

Comparing the tryptophan indole region of the $^1$H-$^{15}$N TROSY-HSQC spectra for the BTK LKD inhibitor bound to Acalabrutinib, Zanubrutinib or Tirabrutinib with that of the apo BTK LKD protein shows that W395 undergoes an upfield shift in the presence of inhibitor (*Figure 3*), suggesting that, like Ibrutinib (*Joseph et al., 2020*) and consistent with the crystal structures, all of these inhibitors stabilize the inactive kinase domain conformation in solution. Interestingly, the Tirabrutinib bound spectrum shows several peaks corresponding to W395, suggesting that the Tirabrutinib-bound BTK kinase domain is likely adopting multiple kinase conformations in solution. Additionally, the magnitude of the upfield shift in W395 resonance is smaller in both the Tirabrutinib and Zanubrutinib spectra

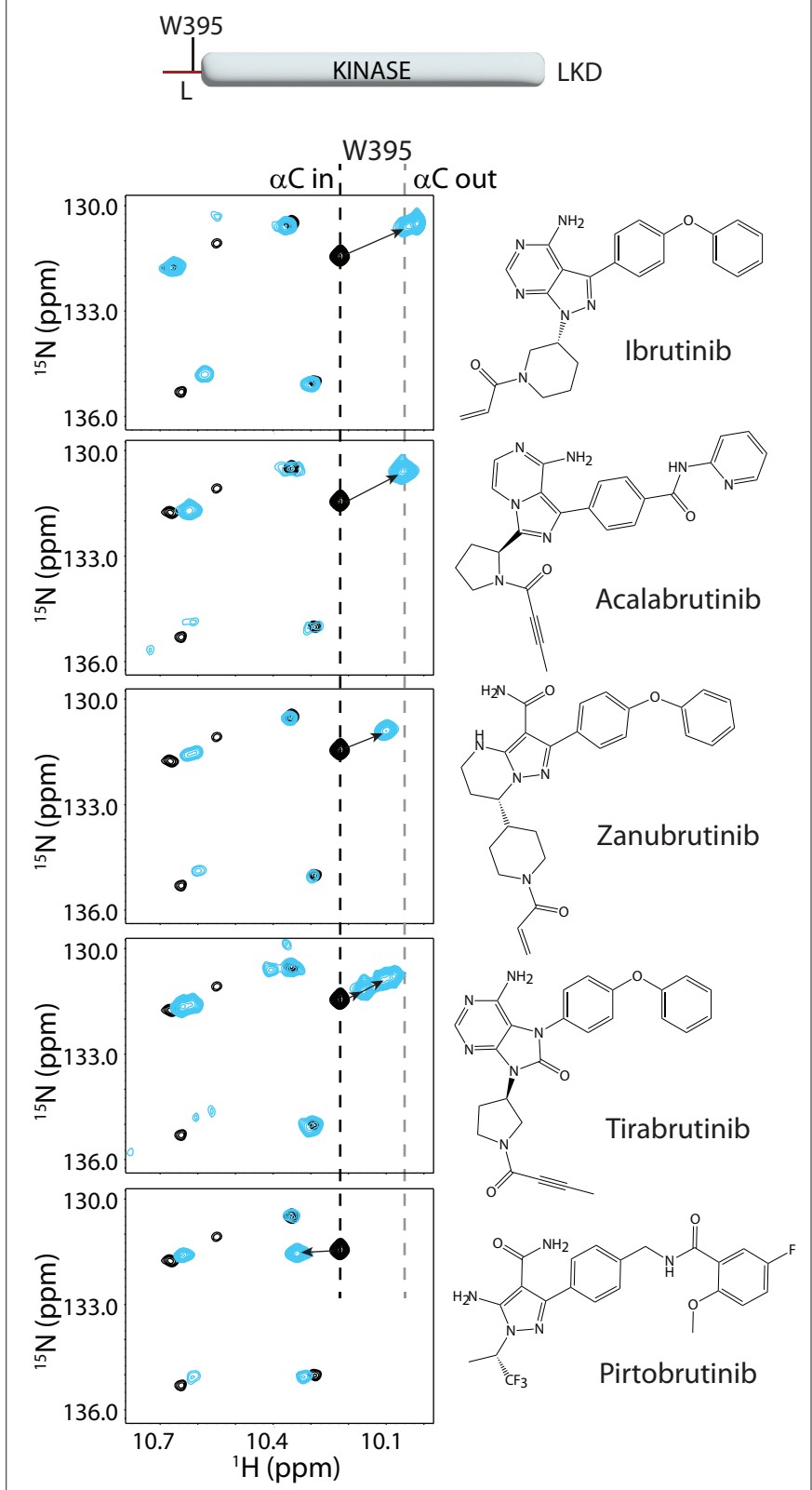

**Figure 3.** Bruton's tyrosine kinase (BTK) inhibitors stabilize the inactive kinase conformation in solution. The tryptophan side chain region of the $^{1}$H-$^{15}$N TROSY HSQC spectra of $^{15}$N-labeled apo BTK linker-kinase domain (black spectrum) overlaid with that of the inhibitor bound spectrum (cyan spectrum). Here and in subsequent figures, the broken black and gray lines indicate the position of the BTK W395 resonance in the active (αC-in) and

*Figure 3 continued on next page*

*Figure 3 continued*

inactive (αC-out) states, respectively as shown in **Figure 2a**. The shift in the BTK W395 resonance upon inhibitor binding is indicated by an arrow in each spectrum. The structures of each inhibitor are shown on the right. The BTK W395 indole NH resonance is in the inactive (αC-out) position in the Ibrutinib (published earlier **Joseph et al., 2020**), Acalabrutinib, Zanubrutinib and Tirabrutinib bound BTK linker-kinase domain (LKD) samples. Multiple peaks corresponding to W395 are seen in the Tirabrutinib bound spectrum suggesting that the kinase adopts multiple conformations in solution. The downfield shift observed in W395 in the Pirtobrutinib bound structure is likely due to local changes in the chemical environment due to the distinct chemical structure of Pirtobrutinib. W395 assignments in the inhibitor-bound spectra were confirmed by acquiring inhibitor-bound spectra with the BTK LKD W395A mutant (see **Figure 3—figure supplement 1**).

The online version of this article includes the following figure supplement(s) for figure 3:

**Figure supplement 1.** Nuclear magnetic resonance (NMR) analysis of inhibitor binding.

(compared to Ibrutinib and Acalabrutinib), which may reflect a relatively larger active state population in these samples.

In contrast to the covalent inhibitors, Pirtobrutinib causes a downfield shift in W395, which might suggest that this inhibitor is stabilizing an active conformation of BTK (**Figure 3**, bottom panel). However, the fluorinated benzene ring in Pirtobrutinib, which is adjacent to the C-helix and W395 in the Pirtobrutinib bound structure, may cause changes in the local environment of W395 and thereby give rise to the unusual chemical shift change. Indeed, HDX-MS data (see below) confirms the stabilization of the inactive kinase conformation by Pirtobrutinib. All W395 assignments in the WT BTK inhibitor bound spectra were confirmed by comparison to the corresponding $^1$H-$^{15}$N TROSY-HSQC spectra of BTK LKD W395A mutant bound to the inhibitors (**Figure 3—figure supplement 1**).

## Probing the effects of inhibitor binding on full-length BTK by HDX-MS

We next evaluated the impact of inhibitor binding on the conformation of full-length BTK by HDX-MS. We have previously shown that HDX-MS can be used to probe conformational changes in full-length BTK that are brought about by inhibitor binding (**Joseph et al., 2020**). The structure of the full-length BTK protein in the autoinhibited conformation was recently solved (**Lin et al., 2024**). However, electron density for both the N-terminal PHTH domain and proline-rich region is missing in this structure, suggesting that the N-terminal region of BTK is dynamic. Indeed, CryoEM, SAXS, and solution data all indicate that the PHTH domain is highly dynamic and likely transiently contacts multiple sites on the core SH3-SH2-kinase domain (**Figure 1b**, **Lin et al., 2024**). Changes in deuterium incorporation must, therefore, be mapped on the crystal structure of the SH3-SH2-kinase fragment of BTK, which adopts a compact inactive autoinhibited conformation (**Figure 1b**, **Wang et al., 2015**). Additionally, our previous work (**Joseph et al., 2020**) has shown that for the majority of the inhibitor complexes studied changes in deuterium incorporation are rarely observed in the PHTH domain suggesting that the PHTH domain dynamics are not affected by active site occupancy. Inhibitor binding typically leads to decreased deuterium incorporation in peptides derived from the kinase active site due to stabilization of this region in the presence of the bound inhibitor (**Joseph et al., 2020**). Additionally, inhibitors such as Ibrutinib that have allosteric effects lead to an increase in deuterium incorporation in peptides derived from the BTK SH3 and SH2 domains as well as the SH2-kinase linker, indicating a shift away from the autoinhibited conformation (**Joseph et al., 2020**). Importantly, HDX-MS provides information about the conformational average adopted by the ensemble and in most cases cannot distinguish between individual intermediate conformations.

Full-length BTK was mixed with Acalabrutinib, Zanubrutinib, Tirabrutinib, or Pirtobrutinib and subjected to HDX-MS analysis. Peptides that could be followed in each of the six experimental conditions (apo, and inhibitor bound) were used for comparison (see **Supplementary file 2**). Intact mass analysis supports a single binding site on BTK for the covalent inhibitors similar to what has been observed previously with Ibrutinib (**Figure 4a**, **Joseph et al., 2020**).

Acalabrutinib, Zanubrutinib, Tirabrutinib, and Pirtobrutinib show decreased deuterium incorporation in peptides derived from the kinase N-lobe and kinase activation loop similar to that observed previously with Ibrutinib (**Figures 4b, c and 5a–e**, **Joseph et al., 2020**). This is consistent with the binding of these inhibitors in the kinase active site and stabilization of the inactive kinase conformation observed by NMR as well as in the crystal structures. Furthermore, we note that each of these

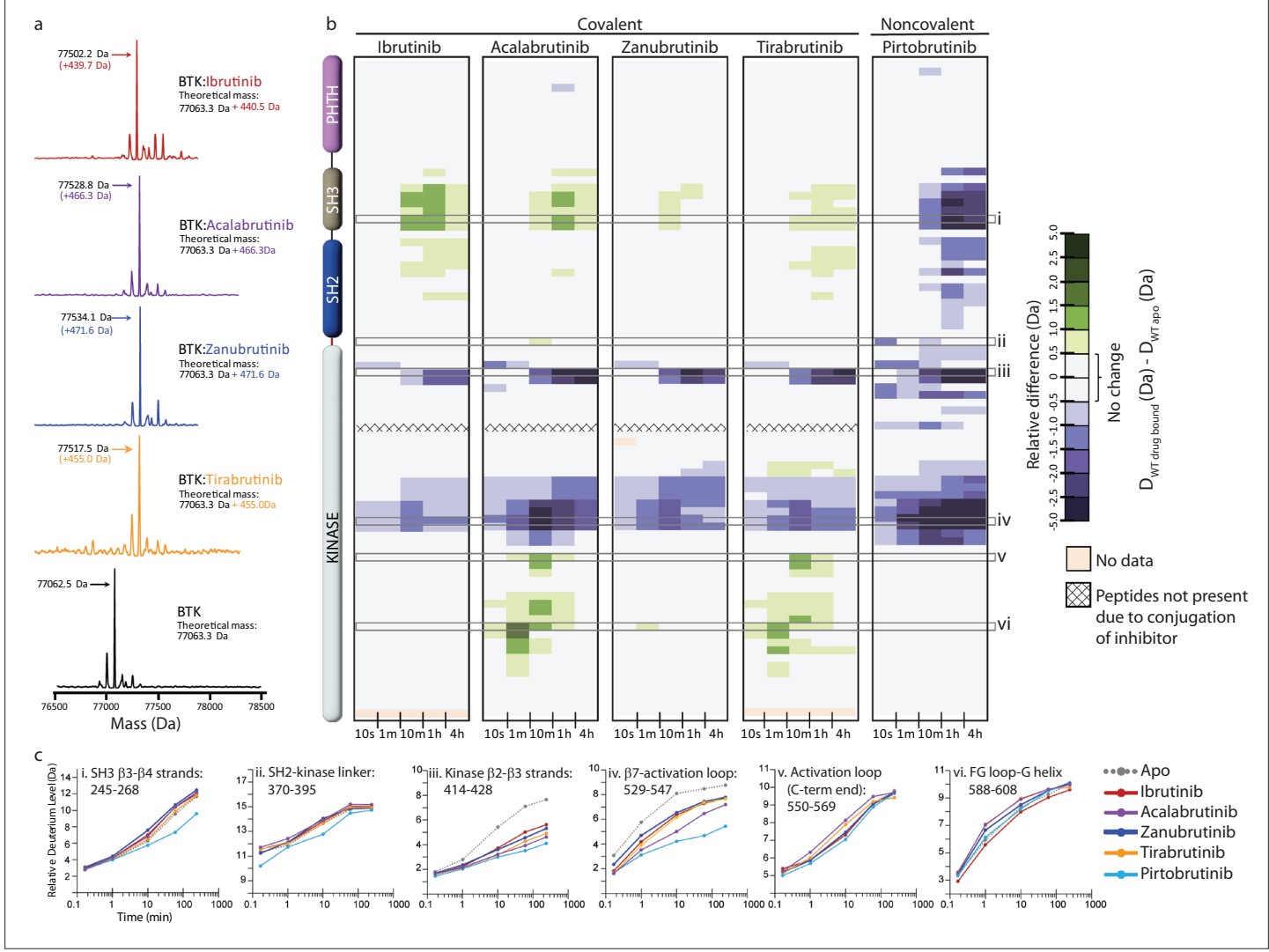

**Figure 4.** Assessing the impact of Bruton's tyrosine kinase (BTK) inhibitors on full-length BTK by hydrogen deuterium exchange mass spectrometry (HDX-MS). (**a**) Intact mass analysis of wild-type full-length (FL) BTK before (bottom spectrum, black) and after 1 hr incubation with a twofold molar excess of covalent BTK inhibitors: Ibrutinib (red), Acalabrutinib (purple), Zanubrutinib (blue), and Tirabrutinib (orange) show a mass increase of one inhibitor molecule. (**b**) Clinically approved BTK inhibitors induce allosteric changes in full-length BTK. Relative deuterium level of peptides in apo BTK was subtracted from the deuterium level of the corresponding peptide from each drug-bound form of BTK ($D_{WT\ drug-bound}$-$D_{WT\ apo}$) and the differences were colored according to the scale shown. In this and subsequent figures, peptic peptides are shown from N- to C-terminus, top to bottom, and the amount of time in deuterium is shown from left to right. The relative difference data shown here represents a curated set of peptides that are coincident across all 6 states (apo and five drug-bound BTK forms). The identification of these chosen peptides, the relative difference values, and the complete data set for each state can be found in the **Supplementary file 2**. The approximate position of the domains of BTK, as described in **Figure 1a**, is shown on the left. Deuterium incorporation curves of selected peptides (indicated with a gray box in panel b and labeled i-vi) from various regions of the protein are shown below. Data for Ibrutinib has been previously published (**Joseph et al., 2020**).

inhibitors induces allosteric effects. Binding of Acalabrutinib, Zanubrutinib, and Tirabrutinib to BTK causes increased deuterium incorporation in peptides derived from the SH3 domain suggesting that the autoinhibited conformation of full-length BTK is destabilized (**Figures 4b, c and 5b–d**). This is similar to what has been previously observed with Ibrutinib (**Joseph et al., 2020**). Additionally, Acalabrutinib and Tirabrutinib showed increased deuterium incorporation in peptides derived from the G-helix of the kinase domain, a region that has been previously identified as the PLCγ substrate docking site (**Figures 4b, c and 5b–d, Xie et al., 2013**). Taken together, Acalabrutinib, Zanubrutinib, and Tirabrutinib binding to BTK leads to hybrid conformations of full-length BTK, where the kinase domain is stabilized in an inactive conformation and the regulatory domains are disrupted from their

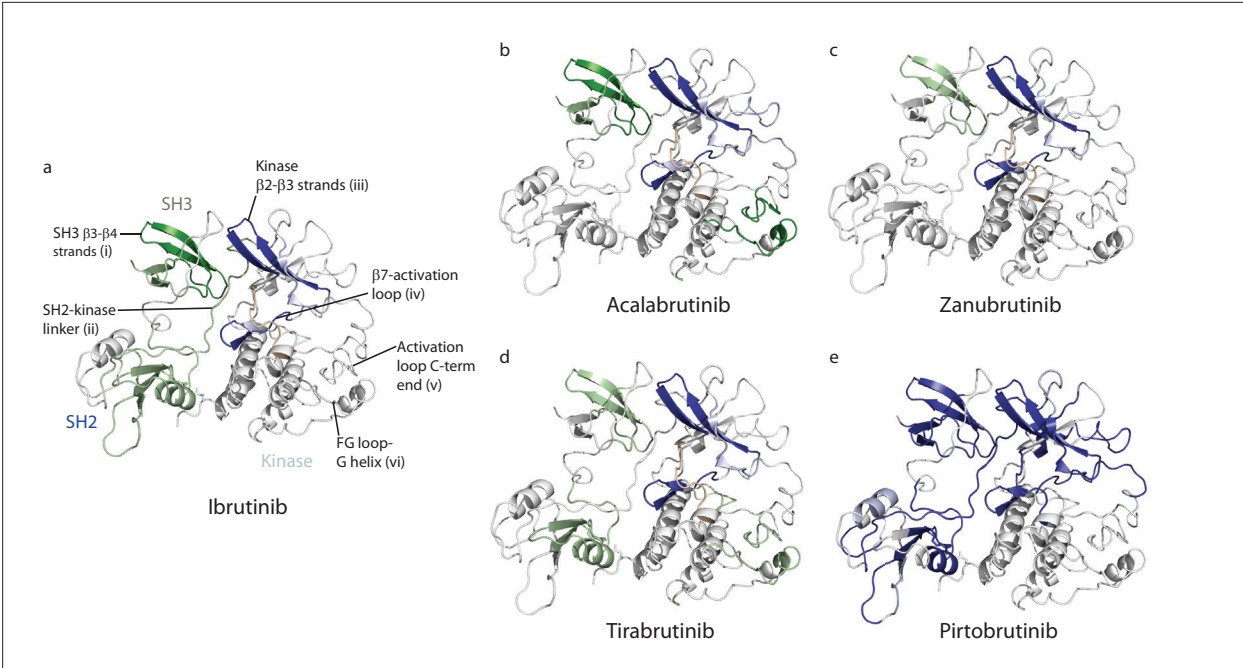

**Figure 5.** Inhibitor binding causes changes in Bruton's tyrosine kinase (BTK). (**a–e**) Mapping the hydrogen deuterium exchange mass spectrometry (HDX-MS) changes induced by each BTK inhibitor on the structure of the BTK SH3-SH2-kinase fragment (PDB ID: 4XI2).Changes mapped onto the structure represent the maximal HDX-MS change that occurred at any time point. Major differences greater than 1.0 Da are shown as dark blue (decrease) or dark green (increase); modest differences between 0.5 Da and 1.0 Da are shown as light blue (decrease) and light green (increase). Localization of the changes in deuterium incorporation was accomplished using overlapping peptides included in the complete peptide data set provided in the **Supplementary file 2**. The location of peptides i – vi from **Figure 4** are indicated in panel a. Data corresponding to Ibrutinib has been previously published (**Joseph et al., 2020**).

autoinhibitory conformation. In stark contrast, Pirtobrutinib binding to BTK shows decreased deuterium incorporation in peptides derived from the SH3 and SH2 domains, suggesting that Pirtobrutinib stabilizes the compact, autoinhibited conformation of full-length BTK (**Figures 4b, c and 5e**). To date, Pirtobrutinib is the only inhibitor that we have tested that uniformly stabilizes both the kinase domain as well as the regulatory domains in the inactive autoinhibited conformation. Thus, each BTK inhibitor causes unique changes in the overall conformation of full-length BTK that is not readily predicted from crystal structures. We next turn our attention to the distinct resistance mutations that arise upon treatment with the different BTK inhibitors.

## Probing the intrinsic effects of the BTK T474I and L528W mutations

BTK resistance mutations can potentially confer a selective advantage to cells in several ways: by increasing the activity of the kinase, by changing the conformation/stability of the kinase which in turn can alter protein-protein interactions, or by disrupting drug binding. To probe the mechanism by which the BTK T474I and L528W mutations confer resistance to Zanubrutinib, Tirabrutinib, and Pirtobrutinib we first tested the impact of each mutation on the catalytic activity of the kinase and then investigated the impact on the overall conformation of the protein by HDX-MS and NMR.

To test the catalytic activity of the BTK mutants we set up in vitro kinase assays using purified full-length WT and mutant BTK proteins. Kinase activity was monitored by following phosphorylation on the activation loop Y551 by western blotting. The BTK T474I mutant shows phosphorylation on Y551, however, it is lower than that of the WT protein (**Figure 6a and b**) suggesting that the T474I mutation reduces the activity of the kinase. In contrast, the BTK L528W mutant is completely inactive with no detectable phosphorylation on Y551 throughout the time course (**Figure 6c and d**). These results are consistent with previous activity reports on both the BTK T474I and L528W mutants (**Wang et al., 2022**; **Handunnetti et al., 2019**; **Johnson et al., 2016**; **Yuan et al., 2022**). Given that neither the BTK T474I, or the L528W mutant had increased catalytic activity, augmenting kinase activity is not the mechanism by which these mutations confer resistance.

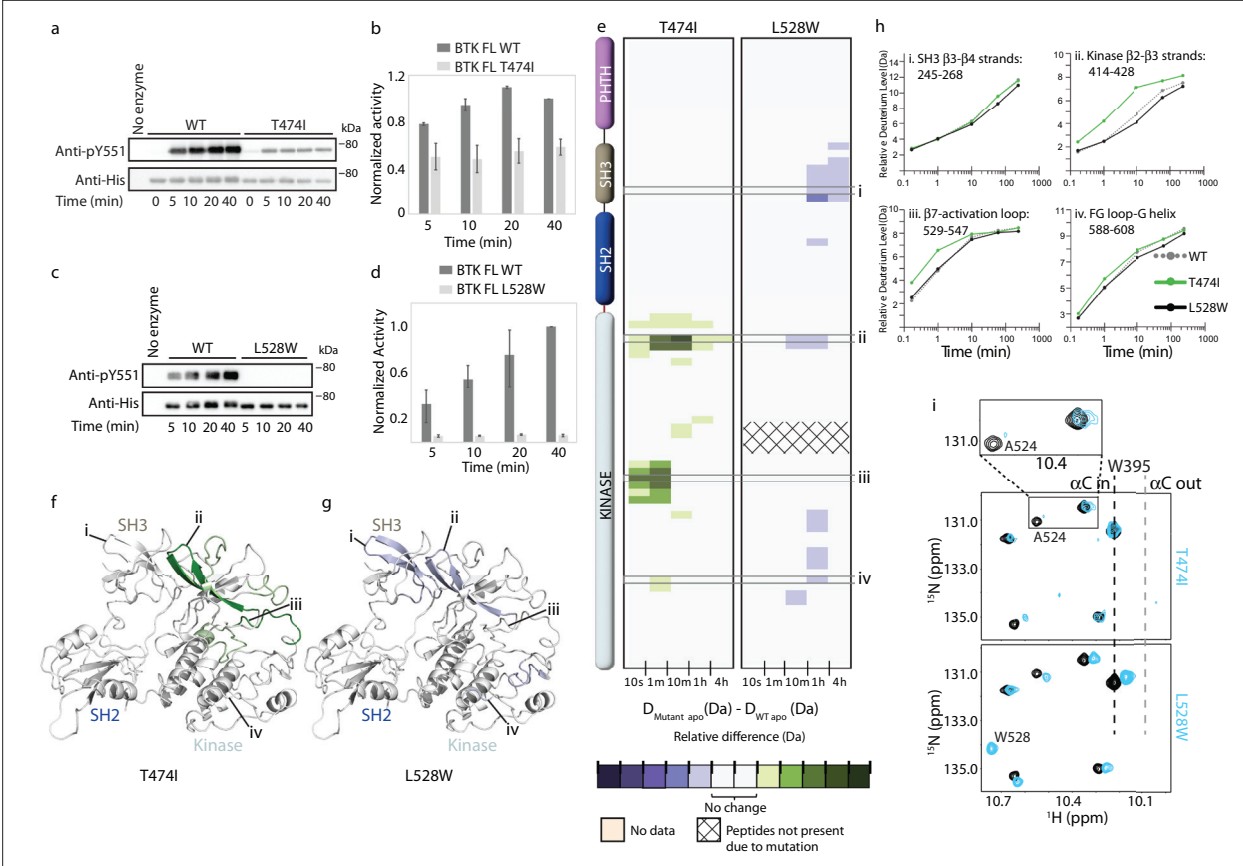

**Figure 6.** Probing the impact of the Bruton's tyrosine kinase (BTK) resistance mutations T474I and L528W on BTK. (**a–d**) Western blot comparing the kinase activity of full-length (FL) BTK wild-type (WT), T474I and L528W mutants. BTK autophosphorylation was monitored using the BTK pY551 antibody and the total protein levels were monitored using the Anti-His antibody. (**b, d**) Histogram quantifying the western blots shown in (**a, c**). The blots were quantified and normalized as described in the Materials and Methods. Data shown are the average of three independent experiments with standard deviations. (**e**) HDX difference data for the BTK T474I and L528W mutants ($D_{Mutant apo}$ -$D_{WT apo}$). Color scale and peptide/time course arrangement are the same as in *Figure 4*. See the *Supplementary file 2* for additional information, including all peptide identifications and deuterium values. (**f, g**) Mapping the mutational induced hydrogen deuterium exchange mass spectrometry (HDX-MS) changes on the structure of the BTK SH3-SH2-kinase fragment. Changes mapped onto the structure represent the maximal HDX-MS change that occurred at any time point. (**h**) Deuterium incorporation curves of selected peptides (indicated with a gray box in panel e and labeled i-iv) from various regions of the protein are shown. (**i**) The tryptophan side chain region of the $^1$H-$^{15}$N TROSY HSQC spectra of $^{15}$N-labeled apo WT BTK linker-kinase domain (black spectrum) overlaid with that of the apo mutant kinase spectrum (cyan spectrum). The boxed region in the T474I spectral overlay is expanded above.

The online version of this article includes the following source data for figure 6:

**Source data 1.** Original unedited files for western blot analysis displayed in *Figure 6a*.

**Source data 2.** *Figure 6a* along with original files for western blots displayed in *Figure 6a* with sample labels.

**Source data 3.** Original unedited files for western blot analysis displayed in *Figure 6c*.

**Source data 4.** *Figure 6c* along with original files for western blots displayed in *Figure 6c* with sample labels.

Resistance mutations can change the conformation of the protein which in turn can alter protein-protein interactions. To test whether the BTK T474I and L528W mutations alter the overall conformation of the protein, we carried out HDX-MS analysis on the apo mutant proteins. Comparing the BTK T474I mutant to WT BTK, the BTK T474I mutant shows increased deuterium incorporation within the kinase domain N-lobe β2-β3 strands and the activation loop, but no changes elsewhere on the protein (*Figure 6e, f and h*). These increased dynamics within the N-lobe of the BTK T474I mutant could potentially alter drug binding to the active site. The BTK L528W mutant on the other hand shows a decrease in deuterium incorporation in the kinase domain and the SH3 domain suggesting that the L528W mutation has a stabilizing effect on the full-length autoinhibited conformation of BTK (*Figure 6e, g and h*).

Additionally, we tested the BTK T474I and L528W mutants by NMR. An overlay of the $^1$H-$^{15}$N TROSY-HSQC spectra of apo BTK LKD T474I with that of apo WT BTK LKD shows some changes in the mutant spectrum (*Figure 6i*). The resonance corresponding to BTK A524 located close (~18 Å) to T474 in the kinase active site shows line broadening in the BTK T474I mutant spectrum suggesting increased dynamics in the mutant. This is consistent with the increased deuterium incorporation observed in the kinase domain for this mutant by HDX-MS. The BTK W395 resonance in the BTK T474I mutant spectrum overlaps with that of the WT suggesting that the mutant kinase domain adopts an active conformation similar to that of the WT protein. The BTK L528W mutant on the other hand shows a small upfield shift in the W395 resonance, suggesting that the BTK L528W mutant kinase domain is shifted towards an inactive conformation compared to the WT protein. This change in the BTK L528W mutant is consistent with the HDX-MS changes for this protein which suggests that the mutation has a stabilizing effect on the compact autoinhibited conformation. Additionally, a new peak is observed as expected in the BTK L528W mutant spectrum consistent with the introduction of an additional tryptophan due to the mutation. Overall, as the BTK T474I and L528W mutations by themselves cause only minor conformational changes in the nucleotide-free apo form, such changes are unlikely to constitute the mechanism by which these mutations confer resistance.

## BTK L528W mutant activates the SRC family kinase HCK

Previous studies have shown that the BTK L528W mutant propagates BCR signaling despite being catalytically dead (*Wang et al., 2022*; *Yuan et al., 2022*). PLCγ phosphorylation and calcium signaling are maintained in cells carrying this BTK mutation (*Wang et al., 2022*; *Yuan et al., 2022*). This suggests that the BTK L528W mutant might recruit other kinases to compensate for its lack of activity. The mechanism of action of another catalytically inactive BTK resistant mutation: BTK C481F/Y, which arises in Ibrutinib treated CLL patients, has been recently reported (*Dhami et al., 2022*). In that work the BTK C481F/Y resistance mutant was shown to bind and activate the SRC family kinase HCK, thereby propagating signals from the BCR signaling pathway (*Dhami et al., 2022*). This activation of HCK by BTK C481F/Y requires phosphorylation on the BTK kinase activation loop, Y551. The phosphorylated Y551 is suggested to bind to the HCK SH2 domain, displacing the autoinhibited conformation of HCK, thereby activating the HCK catalytic function. To determine if a similar mechanism is at work for the catalytically dead BTK L528W mutant, we tested the ability of the BTK L528W mutant to activate HCK in a western blot assay by monitoring the phosphorylation levels on two substrates: PLCγ1 (pY783 phosphorylation) and HCK itself (pY levels). In addition, we also included BTK WT that had been pre-incubated with Zanubrutinib as a control. As shown in *Figure 7a and b*, phosphorylation increased on both PLCγ and HCK in the presence of full-length BTK L528W, suggesting that the catalytically dead BTK mutant is able to activate HCK. Interestingly, Zanubrutinib bound WT BTK also promoted increased phosphorylation levels on both PLCγ1 and HCK suggesting that drug-inactivated BTK also activates HCK. This is distinct from what has been previously observed for the BTK C481Y/F resistant mutation where, compared to WT BTK, the C481Y/F mutant BTK preferentially engages HCK. Nevertheless, since the BTK L528W mutant activates HCK, we next designed experiments to probe the requirements for this activation.

Activation of HCK by BTK C481F/Y requires phosphorylation on the BTK kinase activation loop, Y551. Therefore, we first tested whether phosphorylation on Y551 is required for the activation of HCK by the BTK L528W mutant. While BTK cannot autophosphorylate itself on BTK Y551 in this assay (BTK FL WT is inhibited by Zanubrutinib and BTK FL L528W is catalytically dead), both proteins are able to be phosphorylated by HCK on BTK Y551. Interestingly, phosphorylation on BTK Y551 is lower in the BTK L528W mutant compared to the WT BTK. This is likely due to the BTK kinase activation loop Y551 being less accessible to phosphorylation by HCK in the mutant (which adopts the autoinhibited conformation) compared to WT BTK (*Figure 7a*). Despite the lower pY551 level on BTK FL L528W mutant compared to BTK FL WT, the BTK FL L528W mutant is able to activate HCK similar to the BTK WT protein, suggesting that phosphorylation levels on BTK Y551 do not impact HCK activation. Additionally, we tested whether the BTK L528W/Y551F double mutant is able to activate HCK. The BTK L528W/Y551F double mutant is able to activate HCK similarly to BTK L528W, suggesting that phosphorylation on Y551 is not required for HCK activation by BTK L528W (*Figure 7—figure supplement 1a*). Furthermore, BTK FL L528W increased phosphorylation on HCK and PLCγ1 even in the absence

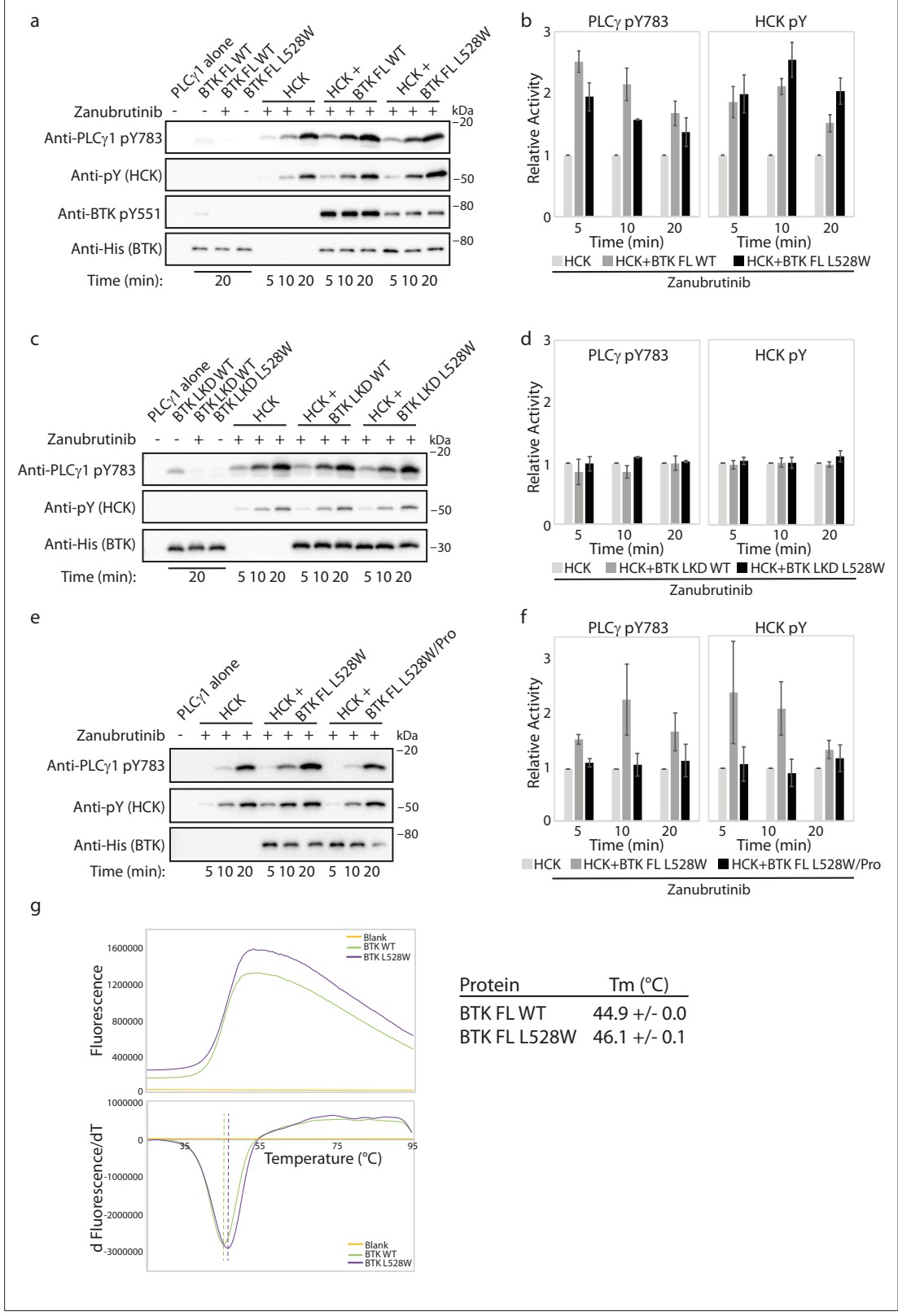

**Figure 7.** The Bruton's tyrosine kinase (BTK) L528W mutant can activate HCK. (**a–f**) Kinase activity of HCK in the presence or absence of full-length BTK L528W mutant was compared in a western blot assay by monitoring PLCγ1 phosphorylation (pY783 antibody) and HCK autophosphorylation (pY antibody). BTK phosphorylation on Y551 was monitored using the Anti-pY551 antibody. Total protein levels were monitored using the Anti-His antibody. Full-length wild-type (WT) BTK preincubated with Zanubrutinib was used as a control. (**b, d, f**) Histogram quantifying the western blots shown in (**a, c,**

*Figure 7 continued on next page*

*Figure 7 continued*

**e**). The blots were quantified and normalized as described in the Materials and methods. Data shown are the average of three independent experiments with standard deviations. (**b, c**) Kinase activity of HCK in the presence or absence of the isolated linker-kinase domain (LKD) fragment of the BTK L528W mutant (**b**) or the full-length proline mutant of BTK L528W (BTK full-length (FL) L528W/Pro: BTK L528W/ P189A/P192A/P203A/P206A, (**c**)) was compared as in (**a**). (**g**) Thermal stability analysis of BTK FL WT and BTK FL L528W. Data shown are the average of three independent experiments with standard deviations.

The online version of this article includes the following source data and figure supplement(s) for figure 7:

**Source data 1.** Original unedited files for western blot analysis displayed in *Figure 7a*.

**Source data 2.** *Figure 7a* with original files for western blots displayed in *Figure 7a* with sample labels.

**Source data 3.** Original unedited files for western blot analysis displayed in *Figure 7c*.

**Source data 4.** *Figure 7c* with original files for western blots displayed in *Figure 7c* with sample labels.

**Source data 5.** Original unedited files for western blot analysis displayed in *Figure 7e*.

**Source data 6.** *Figure 7e* with original files for western blots displayed in *Figure 7e* with sample labels.

**Figure supplement 1.** Bruton's tyrosine kinase (BTK) full-length (FL) L528W is able to activate HCK.

**Figure supplement 1—source data 1.** Original unedited files for western blot analysis displayed in *Figure 7—figure supplement 1a*.

**Figure supplement 1—source data 2.** *Figure 7—figure supplement 1a* with original files for western blots displayed in *Figure 7—figure supplement 1a* with sample labels.

**Figure supplement 1—source data 3.** Original unedited files for western blot analysis displayed in *Figure 7—figure supplement 1b*.

**Figure supplement 1—source data 4.** *Figure 7—figure supplement 1b* with original files for western blots displayed in *Figure 7—figure supplement 1b* with sample labels.

of Zanubrutinib (*Figure 7—figure supplement 1b*) showing that drug interactions are not responsible for the increased phosphorylation on HCK and PLCγ1.

We next tested whether the regulatory domains of BTK were required for HCK activation by the BTK L528W mutant by using the isolated linker-kinase domain of BTK (a construct in which the BTK regulatory domains have been deleted). As shown in *Figure 7c and d*, the isolated linker-kinase domain of BTK L528W does not activate HCK. This suggests that the regulatory domain/s of BTK are required for the activation of HCK and that the isolated linker-kinase domain of BTK alone is insufficient for HCK activation. This again is different from what has been previously reported; the BTK C481Y/F resistant mutant requires the kinase domain for HCK activation (*Dhami et al., 2022*). We, therefore, turned our focus onto the BTK regulatory domains to isolate the specific region/s required for HCK activation by the BTK L528W resistance mutation.

Displacement of the SH3 domain from its autoinhibitory conformation upon binding of a proline-rich ligand is a classic mechanism by which SRC family kinases such as HCK are activated (*Moarefi et al., 1997*). BTK contains such a proline-rich ligand sequence within its regulatory region and so we tested HCK activation by a BTK mutant lacking the proline-rich sequence (BTK L528W/Pro: BTK L528W/ P189A/P192A/P203A/P206A). As shown in *Figure 7e and f*, the BTK L528W proline-depleted mutant does not activate HCK. This data again emphasizes the difference between the mechanism of action of the BTK L528W mutant from the previously characterized BTK C481Y/F mutant. Additionally, this data is consistent with the observation that WT BTK can also activate HCK, as the proline-rich region is present in both WT and mutant L528W BTK. The ability of both WT and L528W mutant BTK to activate HCK, however, suggests that there must be additional reasons for the selection of the BTK L528W resistance mutation in patients.

## BTK L528W is more stable than WT BTK

Changes in protein levels have been shown to cause resistance in several cancers (*Pessoa et al., 2022*). Cellular protein levels can be altered due to changes in protein expression, stability or degradation, or changes at the RNA level (*Pessoa et al., 2022*). An increase in BTK protein levels in vivo could drive protein-protein interactions typically not observed in the WT background. Interestingly, we note that protein yield after bacterial expression and purification of the full-length BTK L528W mutant is higher than that of the WT protein, suggesting that the L528W mutant is more stable than the WT protein. To test whether the BTK L528W mutation has a stabilizing effect on BTK, we measured the melting temperature (Tm) of the BTK L528W mutant and compared it to that of the WT

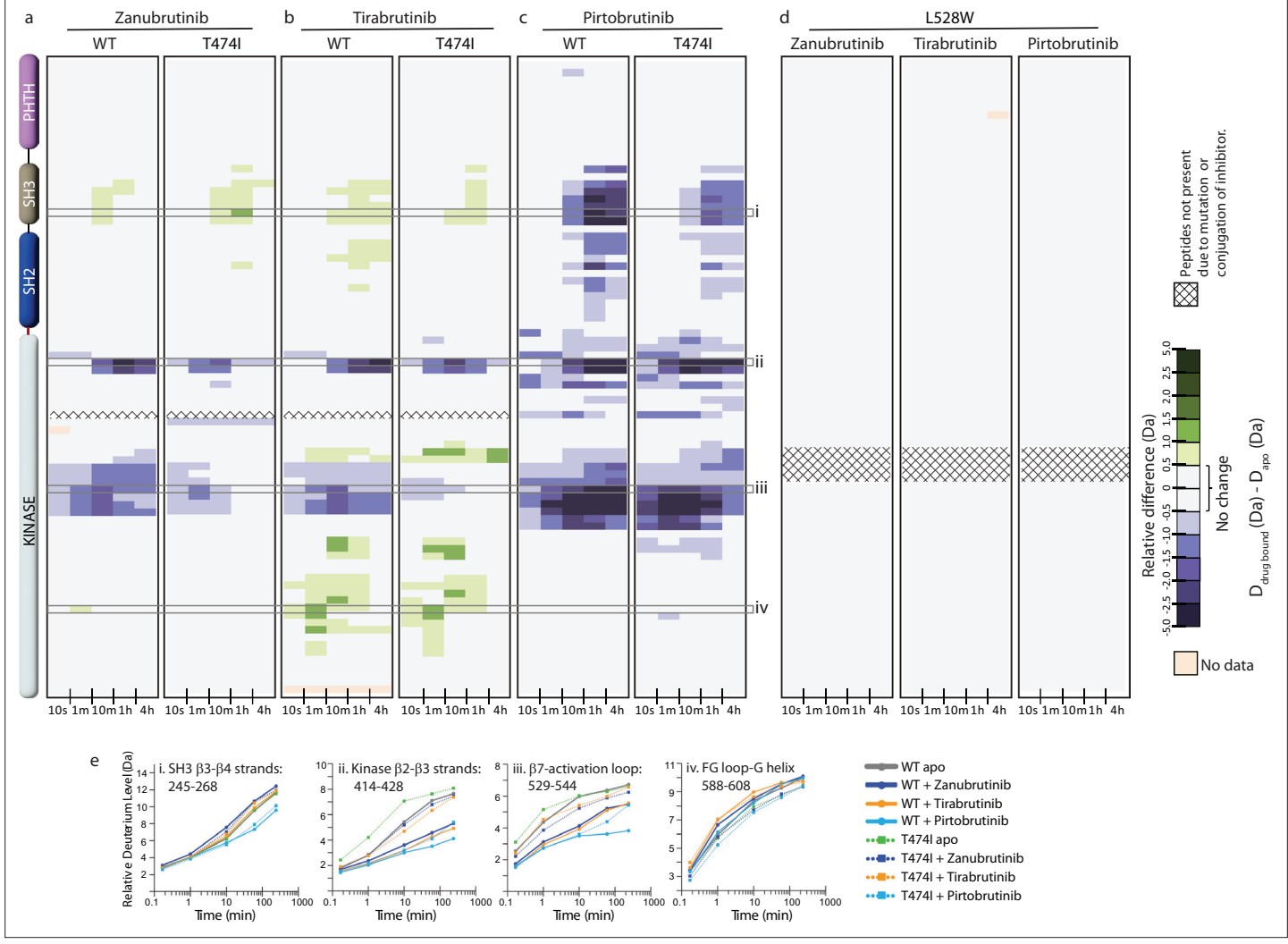

**Figure 8.** Hydrogen deuterium exchange mass spectrometry (HDX-MS) analysis of Bruton's tyrosine kinase (BTK) inhibitor binding to BTK T474I and L528W mutants. (**a–d**) Relative deuterium level of peptides in apo mutant BTK was subtracted from the deuterium level of the corresponding peptide from each inhibitor-bound form of BTK ($D_{drug-bound}-D_{apo}$) and compared to the changes in the wild-type (WT) protein. The differences are colored according to the scale shown. (**e**) Deuterium incorporation curves of selected peptides (indicated with a gray box in panels a-c, and labeled i-iv) are shown.

protein in a thermal shift assay. As shown in *Figure 7g*, the BTK L528W mutant is more stable than the WT protein (a one-degree increase in Tm compared to the WT). The increased stability of the L528W mutant, albeit small, could result in increased protein levels, an increase in the lifetime of the protein, or altered protein trafficking in vivo, all of which would drive the interaction of the mutant protein with HCK, unlike the WT protein.

## HDX-MS reveals that the BTK T474I and L528W mutants show reduced binding to BTK inhibitors

Disruption of inhibitor binding due to mutations is a classic mechanism by which resistance arises (*Bhullar et al., 2018*; *Cohen et al., 2021*). While the loss of inhibitor binding is likely irrelevant to the catalytically dead BTK L528W mutant, we nevertheless evaluated it along with the BTK T474I mutant using HDX-MS and NMR. HDX-MS data show that Tirabrutinib, Zanubrutinib, and Pirtobrutinib induce overall changes in deuterium incorporation for both WT BTK and the T474I mutant (*Figure 8a, b and c*) These data suggest that the inhibitors are still capable of binding to the T474I mutant. However, in the presence of Tirabrutinib, Zanubrutinib, or Pirtobrutinib, the magnitude of the protection in

the kinase domain (N-lobe β2-β3 strands and activation loop) is consistently reduced in drug-bound T474I mutant as compared to the WT protein (*Figure 8a, b, c and e*). The differences in deuterium exchange for drug binding to WT and mutant BTK suggest that the T474I mutation causes a reduction in inhibitor binding. Specifically, the peptides derived from the kinase activation loop in particular do not show as much protection in the mutant relative to the WT suggesting that the inhibitors do not stabilize the activation loop in the inactive conformation upon binding to the T474I mutant. Interestingly, the changes observed in the BTK regulatory domains (SH3 and SH2 domains) upon Zanubrutinib, Tirabrutinib, or Pirtobrutinib binding to WT BTK as well as increased deuterium incorporation in the G-helix of the kinase domain upon Tirabrutinib binding to WT BTK are maintained or reduced in the drug bound BTK T474I mutant. In stark contrast to the BTK T474I mutant, the BTK L528W mutant does not show any change in deuterium incorporation in the presence of Zanubrutinib, Tirabrutinib or Pirtobrutinib, providing strong evidence that the BTK L528W mutant does not bind the inhibitors (*Figure 8d*). Taken together, both the BTK T474I and the BTK L528W mutation impact inhibitor binding. Indeed, these results are consistent with previous studies that show that the BTK T474I and the BTK L528W mutations disrupt binding to Zanubrutinib and Pirtobrutinib (*Wang et al., 2022*). We next tested the impact of the resistance mutations on inhibitor binding by NMR.

## Probing the effects of BTK T474I and L528W mutations on inhibitor binding by NMR

Comparison of the Zanubrutinib bound WT BTK LKD spectrum to that of BTK T474I shows that the BTK T474I mutant undergoes minor changes in the presence of Zanubrutinib, suggesting that the mutant is binding weakly to the inhibitor (*Figure 9a*, left and middle panels). The W395 resonance shows little to no chemical shift change upon the addition of Zanubrutinib to the BTK T474I mutant, suggesting that this drug does not stabilize the inactive conformation of the mutant kinase (*Figure 9a*, middle panel). These results are consistent with HDX-MS changes in the T474I mutant upon Zanubrutinib binding (*Figure 8a*).

Spectral comparison of Tirabrutinib bound to WT and BTK T474I (*Figure 9b*, left and middle panels) shows drug-induced chemical shift changes for both complexes, supporting the conclusion from the HDX-MS data (*Figure 8b*) that the drug binds to both WT and mutant BTK. However, the spectral changes observed in the spectrum of Tirabrutinib bound to BTK T474I are different from that of the WT/Tirabrutinib bound spectrum (*Figure 9b*). Although BTK W395 gives rise to multiple peaks in the spectra of Tirabrutinib bound to both WT and T474I, the W395 resonances in the BTK T474I/Tirabrutinib complex are shifted upfield to a lesser extent compared to the same resonance in the WT/Tirabrutinib spectrum. This suggests that the BTK T474I mutation reduces binding to Tirabrutinib and the inactive kinase domain conformation (αC out) is less populated in the drug-bound mutant protein compared to WT BTK bound to Tirabrutinib. These results are consistent with the HDX-MS results that show reduced protection in peptides corresponding to the kinase-activation loop in the mutant compared to WT due to the lack of stabilization of the kinase inactive conformation upon Tirabrutinib binding.

Spectral overlay of the Pirtobrutinib bound BTK T474I with the apo T474I protein shows pronounced chemical shift changes in the presence of the inhibitor, providing evidence consistent with the HDX-MS data (*Figure 8c*), that Pirtobrutinib binds the BTK T474I mutant (*Figure 9c*, middle). However, unlike Pirtobrutinib bound to WT BTK, multiple W395 peaks are visible in the spectrum of Pirtobrutinib bound to BTK T474I. The resonance frequencies of the additional peaks are shifted toward that of the apo protein suggesting the possibility of fast exchange between inhibitor-bound and unbound states. These observations suggest that the T474I mutation reduces affinity towards Pirtobrutinib and is consistent with previously published SPR binding data (*Wang et al., 2022*).

BTK inhibitors that retain binding to the BTK T474I mutant could serve as potential alternate treatment options for patients with this mutation. Additionally, the lack of or low frequency of the BTK T474I mutation in patients treated with Ibrutinib and Acalabrutinib could be due to the inability of this mutation to disrupt inhibitor binding. We, therefore, tested the ability of the BTK T474I mutant to bind Ibrutinib, Acalabrutinib, and Fenebrutinib (GDC-0853). The NMR data suggest that the BTK T474I mutation does not impact Ibrutinib or Fenebrutinib binding (*Figure 9d and f*), but Acalabrutinib binding is reduced (*Figure 9e*). Taken together, the T474I mutation significantly reduces binding to Zanubrutinib, Tirabrutinib and Pirtobrutinib, retains binding to Ibrutinib and Fenebrutinib, and may

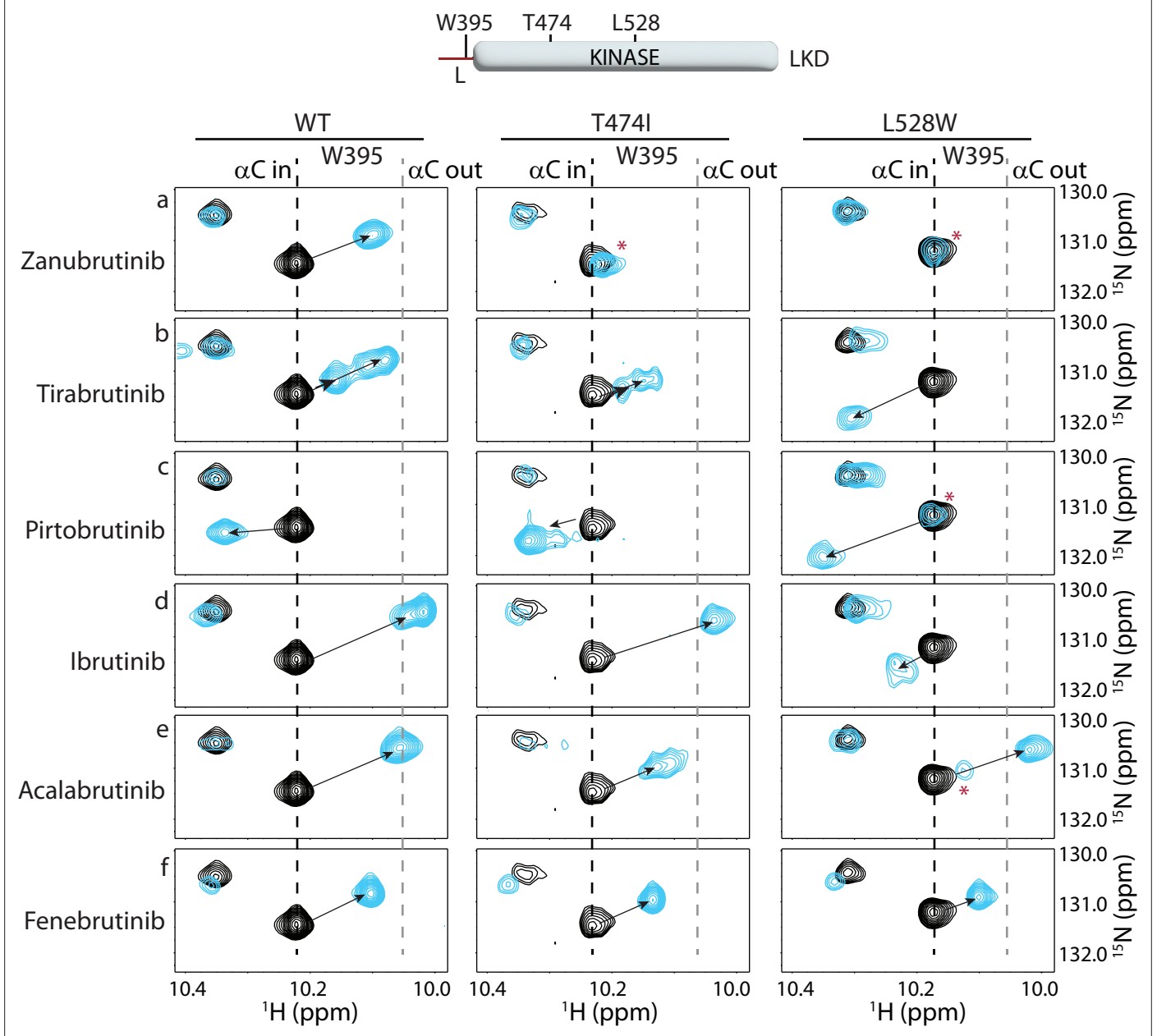

**Figure 9.** Nuclear magnetic resonance (NMR) analysis of Bruton's tyrosine kinase (BTK) inhibitor binding to the BTK T474I and L528W mutants. The tryptophan side chain region of the $^1$H-$^{15}$N TROSY HSQC spectra of $^{15}$N-labeled apo BTK linker-kinase domain (black spectrum) overlaid with that of the inhibitor bound spectrum (cyan spectrum). The broken black and gray lines indicate the position of the BTK W395 resonance and have been described earlier in *Figure 3*. The shift in the BTK W395 indole NH resonance upon inhibitor binding is indicated by an arrow in each spectrum. The red asterisks indicate the presence of an unbound kinase domain in the inhibitor-bound NMR sample.

The online version of this article includes the following figure supplement(s) for figure 9:

**Figure supplement 1.** Overlay of the $^1$H-$^{15}$N TROSY HSQC spectra of $^{15}$N-labeled apo Bruton's tyrosine kinase (BTK) linker-kinase domain (LKD) L528W (black spectrum) with that of the Pirtobrutinib bound BTK LKD L528W spectrum (cyan spectrum).

have an adverse effect on Acalabrutinib binding. Given that the BTK T474I mutant retains activity (albeit reduced activity), disruption of drug binding is a possible mechanism by which this mutation escapes inhibition.

Spectral overlays of the BTK L528W mutant with and without Zanubrutinib show no chemical shift changes (*Figure 9a*, right panel) suggesting that the mutation completely disrupts inhibitor binding in complete agreement with the HDX-MS data (*Figure 8d*). Tirabrutinib does show chemical shift

changes, but the changes are markedly different from that observed in the WT protein (*Figure 9b*, right). In the L528W spectrum in the presence of Tirabrutinib, W395 shifts in the opposite downfield direction compared to the WT spectrum suggesting that Tirabrutinib could be binding to the BTK L528W in a different orientation. Alternatively, the distinct chemical shift change could be due to the mutated L528W residue altering the local chemical environment within the kinase.

The Pirtobrutinib-bound BTK L528W spectrum (*Figure 9c*, *Figure 9—figure supplement 1*) shows two resonance positions, one of which overlaps with the resonance in the apo protein and the other that corresponds to that of the mutant protein bound to Pirtobrutinib. This data suggests a mixture of inhibitor-bound and unbound BTK kinase domains in solution, likely due to a reduction in Pirtobrutinib affinity caused by the L528W mutation. Although the L528W mutation reduces binding to both Tirabrutinib and Pirtobrutinib, the NMR data suggests that it retains partial binding, unlike the HDX-MS data that suggests complete disruption of binding. The higher inhibitor concentrations used in the NMR experiments compared to the HDX-MS experiments likely explain this discrepancy. Interestingly, we note similarities between the BTK L528W Pirtobrutinib bound spectrum and that of the Tirabrutinib bound spectrum, suggesting that the BTK kinase domain adopts similar conformations in solution when bound to these different drugs. Additionally, the BTK L528W mutant retains binding to Fenebrutinib, however Ibrutinib, and Acalabrutinib binding are disrupted (*Figure 9d–f*, right). Taken together, the BTK L528W mutation significantly disrupts Zanubrutinib, Tirabrutinib, and Pirtobrutinib binding. However, drugs based on Fenebrutinib could be developed to treat patients carrying this mutation (see discussion below).

## Discussion

Binding of BTK active site inhibitors can have long-range effects on the protein. Here, we build on earlier work to show that three of the clinically approved BTKi (Acalabrutinib, Zanubrutinib, and Tirabrutinib) shift the conformational ensemble of full-length BTK, destabilizing the autoinhibited conformation of the SH3 and SH2 domains to varying degrees (*Figures 4 and 5*). In marked contrast, Pirtobrutinib led to the stabilization of the compact autoinhibited conformation of full-length BTK. The exposure or stabilization of the regulatory domains of BTK by active site BTK inhibitors can alter the interaction of BTK with ligands in vivo. Indeed, BTK inhibitors have been shown to differ in their effectiveness at terminating signals downstream of the Fc receptor *versus* the B-cell receptor (*Bender et al., 2017*; *Li et al., 2024*). As more BTKi become available, a molecular-level understanding of the interaction of the inhibitor with full-length BTK will aid the interpretation of the efficacy of different BTK inhibitors in treating disease states driven by different signaling pathways.

The development of specific resistance mutations in patients treated with different BTKi is intriguing. Covalent BTKi rely on BTK C481 within the kinase active site. The C481S mutation prevents the covalent attachment of these inhibitors to BTK and converts the mode of binding of these drugs to that of a reversible inhibitor. Importantly, we and others have shown that the BTK C481S mutation does not prevent the binding of Ibrutinib to the BTK kinase domain (*Joseph et al., 2020*; *Wang et al., 2022*). In fact, the BTK C481S mutant binds to Ibrutinib just as well as the WT protein in vitro under equilibrium conditions where drug concentrations do not vary over time (*Joseph et al., 2020*). However, in patients, drug concentrations change with time; inhibitor concentrations peak rapidly after intake followed by a decrease as the drug is cleared (represented by the half-life of the drug). The occupancy

**Table 1.** Half-life of clinically approved Bruton's tyrosine kinase (BTK) inhibitors: Ibrutinib (*Advani et al., 2013*), Acalabrutinib (*Podoll et al., 2019*), Zanubrutinib (*Tam et al., 2019*), Tirabrutinib (*Walter et al., 2016*), and Pirtobrutinib (*Mato et al., 2021*).

| Inhibitor | Half-life (h) |
| --- | --- |
| Ibrutinib | 2–3 |
| Acalabrutinib | 0.6–2.8 |
| Zanubrutinib | 4 |
| Tirabrutinib | 6.5–8 |
| Pirtobrutinib | 20 |

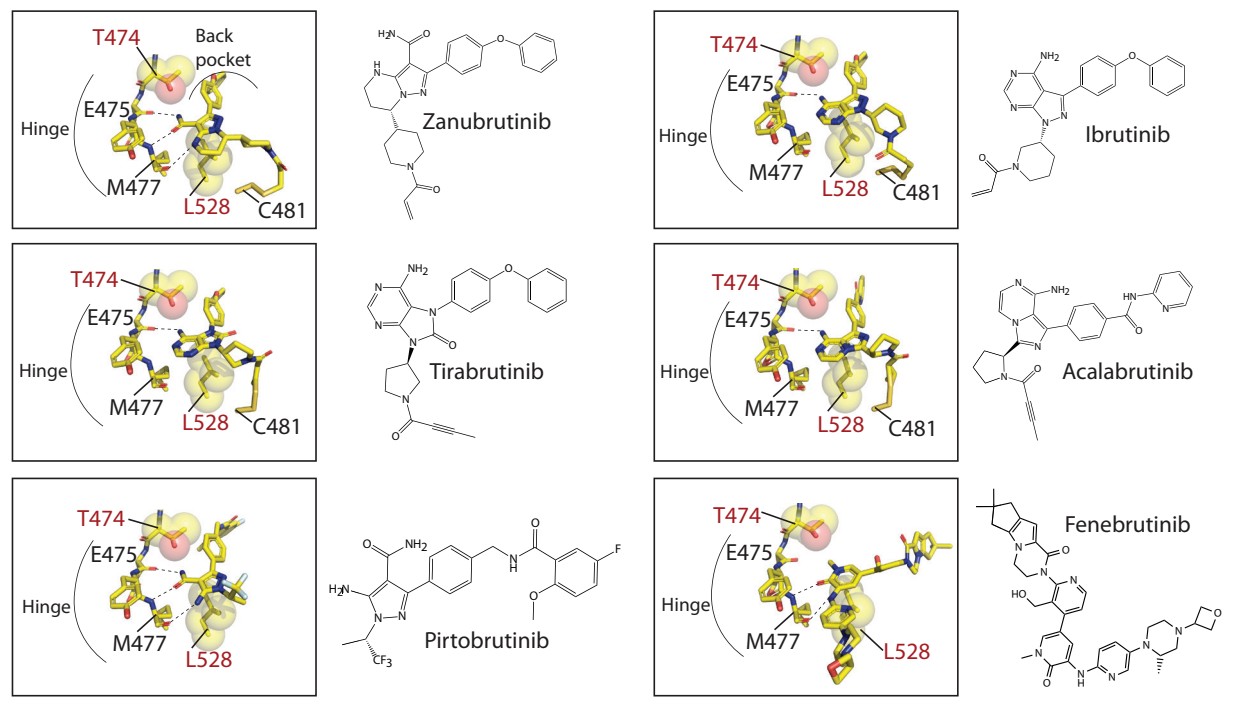

**Figure 10.** Close-up of Bruton's tyrosine kinase (BTK) inhibitors bound to the BTK kinase active site. The covalent BTK inhibitors (Zanubrutinib, Tirabrutinib, Acalabrutinib, and Ibrutinib, represented as sticks) are bound to BTK C481. All BTK inhibitors, covalent and non-covalent (Pirtobrutinib and Fenebrutinib) interact with the hinge region of the kinase and with the exception of Fenebrutinib, extend into the back pocket of the kinase. Hydrogen bonds between the inhibitor and hinge region are shown as dotted lines. The binding of these BTK inhibitors are incompatible with nucleotide binding. The location of BTK T474 and L528, residues that are frequently mutated in chronic lymphocytic leukemia (CLL) patients are shown as sticks and spheres. BTK T474 is adjacent to the hinge region and BTK L528 is at the 'base' of the kinase active site.

of BTK C481S protein by these covalent inhibitors in patients is, therefore, dictated by the availability of these covalent inhibitors over time. Interestingly, Ibrutinib and Acalabrutinib have short half-lives compared to Zanubrutinib and Tirabrutinib (*Table 1*). The short half-lives of Ibrutinib and Acalabrutinib suggest that the BTK C481S mutant is likely to be unoccupied when inhibitor concentrations fall, thereby allowing the BTK C481S mutant to escape from inhibition. Conversely, the longer half-life of Zanubrutinib and Tirabrutinib in patients suggests that the BTK C481S mutant is likely to remain occupied (inhibited) for a longer period of time. The C481S mutation, therefore, does not offer a selective advantage as a resistance mutation in Zanubrutinib and Tirabrutinib treated patients and likely explains the lower frequency of occurrence of this mutation in these patients compared to Ibrutinib and Acalabrutinib treated patients. Taken together, the BTK C481S mutant can escape covalent inhibition at low drug concentrations in vivo and is, therefore, predicted to arise in patients treated with BTK covalent inhibitors that have short half-lives.

The T474I mutation arises frequently in Tirabrutinib and Pirtobrutinib-treated CLL patients (*Figure 1d*). Our HDX-MS and NMR binding studies show that the T474I mutation disrupts binding to both inhibitors. Reduced binding of BTK T474I mutant to Tirabrutinib and Pirtobrutinib, along with the partial catalytic activity retained by this mutant, would allow BTK T474I mutant to escape inhibition and maintain BCR signaling. This gives a selective advantage to the cells that carry this mutation. Additionally, our NMR binding studies show that the BTK T474I mutation does not disrupt the binding of Ibrutinib. This is consistent with the low occurrence of this mutation in Ibrutinib-treated CLL patients. While the BTK T474I mutation does seem to disrupt binding to Acalabrutinib, the lower activity of the T474I mutant (compared to the BTK C481S mutant) may explain the higher prevalence of the BTK C481S mutation (compared to T474I) in Acalabrutinib treated patients.

Analysis of the crystal structures of BTK bound to inhibitors provides clues as to why some BTK inhibitors are more susceptible to loss of binding due to the BTK T474I and L528W mutations. All BTK inhibitors evaluated in this study occupy the kinase active site, contact the kinase hinge region

(BTK residues E475-M477), and (with the exception of Fenebrutinib) extend into the back pocket of the kinase (*Figure 10*). BTK T474 is adjacent to the kinase hinge region and BTK L528 occupies the 'base' of the kinase active site. Given the central location of both these BTK residues, the introduction of the bulky T474I and L528W mutations are likely to impact multiple inhibitor interactions. Zanubrutinib and Pirtobrutinib in particular, make extensive contacts with the kinase hinge region with three hydrogen bonds to BTK E475 and M477 backbone atoms. In contrast, Ibrutinib makes only a single hydrogen bond with BTK E475 backbone. This likely explains why the BTK T474I and L528W mutations disrupt Zanubrutinib and Pirtobrutinib binding compared to Ibrutinib. The BTK inhibitor Fenebrutinib also seems to be less impacted by the BTK T474I and L528W mutations. The BTK T474I and L528W mutations are likely accommodated as Fenebrutinib does not occupy the 'back pocket' of the kinase active site. While these BTK WT inhibitor-bound structures provide clues as to why the efficacy of certain BTK inhibitors are more impacted by the BTK T474I and L528W mutations, some questions remain. Tirabrutinib for example makes a single hydrogen bond with the hinge region, however, it is severely impacted by the BTK T474I and L528W mutations. These mutations likely disrupt additional interactions that Tirabrutinib makes with the kinase such as the glycine loop of the kinase. Crystal structures of inhibitors bound to the BTK mutants are needed to conclusively evaluate their effects.

## Ideas and speculation

Successful treatment of CLL has relied on the inhibition of BTK catalytic activity to curb BCR signaling. Inhibitor-bound, catalytically-inactive BTK is incapable of propagating BCR signaling. Paradoxically, the catalytically inactive BTK L528W mutant is able to propagate BCR signals (*Wang et al., 2022*; *Yuan et al., 2022*). A similar catalytically inactive BTK mutant, the BTK C481Y/F resistance mutation also arises in Ibrutinib and Acalabrutinib-treated patients, albeit at a low frequency (*Dhami et al., 2022*). Since these BTK mutants are catalytically dead, other kinases must compensate for the absence of BTK activity to account for the intact BCR signaling in these patients. Exogenous kinases do not appear to compensate for WT BTK that is inactivated by BTK inhibitors, and so there must be additional differences between the catalytically dead (BTK L528W and BTK C481Y/F) mutant kinases and inhibitor-bound WT BTK that is rendered catalytically inactive by virtue of drug binding. Our HDX-MS and NMR data show that BTK L528W mutation does not drastically change the conformation of BTK. However, this mutation does increase the stability of BTK compared to the WT protein. This increased stability could alter protein levels or change the trafficking/localization of the mutant protein in vivo which in turn could alter protein-protein interactions that drive activation of an alternate kinase such as HCK in vivo. Comparison of BTK protein levels in patients pre- and post-development of resistance will be needed to test this hypothesis. Alternatively, the BTK L528W mutation may require an additional change/s in the cell that allows for this dead kinase to recruit other kinases such as HCK to propagate BCR signaling. This requirement for additional changes may explain the low frequency of the L528W mutation in Ibrutinib and Acalabrutinib-treated patients; the single amino acid change, C481S, combined with the rapid clearance of the drug seems sufficient to drive resistance. Finally, the absence of the BTK L528W resistance mutation in Ibrutinib-treated CLL patients could be due to the promiscuity of Ibrutinib; candidate compensatory kinases may be inactivated by the drug (*Yang et al., 2016*). The increased specificity of Zanubrutinib, Tirabrutinib, and Pirtobrutinib on the other hand could allow for the utilization of compensatory kinases such as HCK by BTK L528W in order to propagate BCR signaling.

Understanding the mechanism/s by which resistance mutations evade inhibition allows for counter strategies to be devised (*Skånland and Mato, 2021*; *Smith and Burger, 2021*; *Chirino et al., 2023*). The reversible BTK inhibitor Pirtobrutinib is able to inhibit the BTK C481S mutant due to its long serum half-life and has been suggested as a treatment option for patients who develop this mutation (*Aslan et al., 2022*). Alternatively, drug occupancy in the context of the BTK C481S resistance mutant could be increased by administering a twice-daily dose of the inhibitor as opposed to the single daily dose possibly circumventing their short half-lives (*Skånland and Tjønnfjord, 2022*). This increased exposure time to the covalent inhibitor could cause increased side effects so such risks should be considered before altering the dosage regimen. Alternatively, Tirabrutinib, which exhibits the longest half-life among the available covalent BTK inhibitors, could be used to slow the progression of disease in patients that develop the C481S resistance mutation.

The BTK T474I mutation disrupts binding to Zanubrutinib, Tirabrutinib, Pirtobrutinib, and Acalabrutinib (*Figures 8 and 9*) but retains binding to Ibrutinib and Fenebrutinib creating possible options for patients that develop the BTK T474I resistance mutation. The BTK L528W mutant is catalytically inactive, hence treatment with any BTK active site inhibitor is futile. Previous studies have shown that the similarly inactive BTK C481Y/F mutant can be suppressed by the use of PROTACS to induce the degradation of the full-length protein (*Dhami et al., 2022*). Importantly, our data indicate that although the BTK L528W mutation disrupted binding to most BTK inhibitors it retains binding to Fenebrutinib (*Figure 9*), suggesting that PROTACS based on Fenebrutinib or similar backbones could be developed to treat this resistance mutation (*Zorba et al., 2018*). Alternate treatment approaches such as targeting HCK or other BCR signaling proteins such as BCL-2 (*Molica, 2020*) could also be used to counter this resistance mutation. As additional patient data becomes available, the patterns of resistance mutations for different inhibitors will become clearer and strategies to circumvent resistance using existing inhibitors should improve. As for many areas of medicine, the treatment landscape should benefit from the ongoing era of personalized medicine.

## Materials and methods
### Constructs and reagents
The bacterial expression constructs for murine BTK LKD and full-length (FL) have been described previously (*Joseph et al., 2017*). All BTK constructs carry the solubilizing Y617P mutation for bacterial expression (*Joseph et al., 2017*). All mutations were made using the site-directed mutagenesis kit (Agilent), and the sequences of all constructs were confirmed by sequencing at the Iowa State University DNA facility. Acalabrutinib, Zanubrutinib, Tirabrutinib, and Pirtobrutinib were purchased from MedChem Express. Ibrutinib was purchased from Selleckchem. The pCDF-1 Duet HCK SH3-SH2-Kinase domain/YopH construct with a 'YEEI' tail, a kind gift from Dr. Tom Smithgall, was mutated to convert the 'YEEI' tail residues to that of the WT protein (YQQQ). The pGEX PLCγ1 cSH2-linker Y771F/Y775F protein substrate construct has been described previously (*Joseph et al., 2007*).

### Protein expression and purification
Expression and purification of the PLCγ1 cSH2-linker Y771F/Y775F protein substrate has been described previously (*Joseph et al., 2007*). HCK, Full-length BTK, and BTK linker-kinase domain (WT and mutants) were produced by co-expressing with YopH in BL21(DE3) (Millipore Sigma) or BL21-Gold(DE3) cells (Agilent Technologies) as described previously (*Joseph et al., 2017*). Briefly, the culture was grown at 37 °C to an O.D. 600 nm of 0.6–0.8. The temperature of the culture was lowered to 18 °C and then induced with either 1.0 mM IPTG for BTK LKD constructs or 0.1 mM IPTG for BTK full-length and HCK. The culture was harvested 24 hr after induction and the pellets were resuspended in lysis buffer (50 mM $KH_2PO_4$, pH 8.0, 150 mM NaCl, 20 mM imidazole, and 0.5 mg/ml lysozyme) and stored at –80 °C. Cells were lysed by thawing and the action of lysozyme, and 3000 U DNAse I (Sigma) and 1 mM PMSF were added to the lysate, incubated at RT for 20 min, and then spun at 16,000 rpm for 1 hr at 4 °C. Glycerol was added to the supernatant to a final concentration of 10% and was then incubated with Ni-NTA resin (QIAGEN) for 2 hr, washed with Tris pH 8.0, 75 mM NaCl, 40 mM imidazole, and eluted in 20 mM Tris pH 8.0, 150 mM NaCl, 250 mM Imidazole, and 10% glycerol. Eluted protein was flash-frozen in liquid nitrogen and stored at –80 °C. The proteins were concentrated and further purified by size exclusion chromatography (Hiload Superdex 26/60 200 pg or Hiload Superdex 26/60 75 pg, GE Healthcare). The fractions containing pure protein were pooled, concentrated, snap-frozen, and stored at –80 ° C. The final buffer consists of 20 mM Tris pH 8.0, 150 mM Sodium chloride, 0.02% Sodium azide, and 10% glycerol. Initial phosphorylation levels of all purified BTK and HCK proteins used in this study are below western immuno-detection.

### NMR
Uniformly $^{15}N$ labeled BTK samples were produced as described earlier by growth in modified M9 minimal media containing $^{15}N$ ammonium chloride (1 g/L, Cambridge Isotope Laboratories, Inc) as the sole source of nitrogen (*Joseph et al., 2017*). The final NMR sample buffer consists of 20 mM Tris, 150 mM Sodium chloride, 10% glycerol, and 0.02% Sodium azide at pH 8.0. All NMR spectra were collected at 298 K on a Bruker AVIII HD 800 spectrometer equipped with a 5 mm HCN z-gradient

cryoprobe operating at a [1]H frequency of 800.37. NMR samples with inhibitors consisted of 150 μM [15]N labeled BTK, mixed with 200 μM inhibitor in 2% DMSO. All data were analyzed using NMRViewJ (*Johnson and Blevins, 1994*).

## HDX-MS

General procedures for HDX-MS of BTK have been described in detail previously (*Joseph et al., 2017*). Details specific to experiments conducted here are provided in the *Supplementary file 2* in the format recommended (*Masson et al., 2019*) for HDX-MS experimental descriptions. All HDX-MS data have been deposited to the ProteomeXchange Consortium via the PRIDE (*Vizcaíno et al., 2016*) partner repository with the dataset identifier PXD047865. Briefly, prior to continuous labeling HDX experiments, purified BTK full-length wild-type, T474I or L528W (20 μM) and inhibitor (40 μM), (20 mM Tris pH 8.0, 150 mM NaCl, 10% glycerol, 2% DMSO) were allowed to interact at 21 °C for 1 hr. After the binding reactions, both the free kinase and kinase bound to the inhibitor were placed on ice prior to deuterium labeling. Deuterium labeling proceeded for the times described using labeling buffer, and labeling was stopped with an equal volume of quench buffer at 0 °C (details in *Supplementary file 2*). Quenched samples were immediately analyzed using a Waters nanoACQUITY with HDX technology using online pepsin digestion with a Waters Enzymate immobilized pepsin column and UPLC separation of the resulting peptic peptides. Mass spectra were acquired using a Waters Synapt HDMS[E] mass spectrometer. Peptides generated from online pepsin digestion were identified with Waters Protein Lynx Global Server 3.0 using a separate unlabeled protein that was prepared in the same manner as protein labeled with deuterium. Deuterium incorporation was quantified using Waters DynamX 3.0. Deuterium levels for each peptide were calculated by subtracting the average mass of the undeuterated control sample from that of the deuterium-labeled sample; the data were not corrected for back exchange and are, therefore, reported as relative (*Wales and Engen, 2006*). Vertical difference maps in *Figures 4, 6 and 8* do not represent a linear sequence of non-overlapping peptides. All coincident and overlapping peptides for comparisons in each figure are provided in figure identified tabs of the *Supplementary file 2*.

## Activity assays

In vitro kinase assays were performed as described previously (*Joseph et al., 2017*). Briefly, 1 μM BTK FL, BTK FL T474I, or BTK FL L528W proteins were incubated in a kinase assay buffer (50 mM Hepes pH 7.0, 10 mM MgCl$_2$, 1 mM DTT, 5% glycerol, 1 mM Pefabloc, and 200 μM ATP) at room temperature for varying time. The reactions were stopped by the addition of SDS-PAGE loading buffer and the samples were boiled, separated by SDS−PAGE, and Western blotted with the anti-BTK pY551 antibody (BD Pharmingen) or anti-His antibody (EMD Millipore) as described previously (*Joseph et al., 2017*). The bands were quantified using the ChemiDoc (Biorad) gel imaging system. The phosphorylation levels (Anti-BTK pY551 blot) were normalized to the total protein level (Anti-His blot). The BTK FL value at 40 min was set to 1 and compared to BTK FL T474I or BTK FL L528W. The HCK activation experiments were performed in the absence of Zanubrutinib or by preincubating 0.2 μM of BTK WT, L528W, or L528W/Y551F proteins with 0.5 μM Zanubrutinib for 15 min at RT. The reaction was initiated with the addition of 0.2 μM HCK in a kinase assay buffer (50 mM Hepes pH 7.0, 10 mM MgCl$_2$, 1 mM DTT, 5% glycerol, 1 mM Pefabloc, 30 μM PLCγ1 cSH2 linker Y771F/Y775F substrate and 200 μM ATP). Time points were taken at 5, 10, and 15 min. The reactions were stopped by the addition of SDS-PAGE loading buffer and the samples were boiled, separated by SDS−PAGE, and Western blotted with the anti-PLCγ1-pY783 antibody (Cell Signaling) or anti-pY antibody (4G10, Millipore Sigma) or anti-His antibody (EMD Millipore) as described previously (*Joseph et al., 2017*). The bands were quantified using the ChemiDoc (Biorad) gel imaging system. The HCK alone value was set to 1 and compared to HCK +BTK WT/L528W. Initial phosphorylation levels of BTK, prior to the start of the activity assay were undetectable by western immuno-detection.

## Thermal shift assays

BTK FL WT or mutant at 5 μM was incubated with 10 X final concentration of SYPRO Orange dye (Thermo Fisher Scientific) in a total volume of 20 μL Tris buffer (20 mM Tris, 150 mM Sodium chloride, 10% glycerol, pH 8.0). Thermal shift assays were performed on the StepOnePlus RT-PCR machine in the Iowa State University DNA Facility. The temperature was raised from 25–95 °C and measurements

were recorded with every 0.5 °C increment in temperature. The derivative plot of the data was analyzed for the melting temperature (Tm). Assays were performed thrice and the average Tm was calculated.

## Materials availability statement

Expression constructs for Btk and PLCγ1 may be obtained by contacting the corresponding author.

## Acknowledgements

This work is supported by a grant from the National Institutes of Health (National Institute of Allergy and Infectious Diseases, AI43957) to AHA, JRE, and TEW, and by funds from the Scott-Waudby Trust, the Hope Against Cancer charity, Cancer Research UK in conjunction with the UK Department of Health on an Experimental Cancer Medicine Centre grant [C10604/A25151] to SJ, RGB, and MJSD. Research at the University of Leicester was carried out at the National Institute for Health and Care Research (NIHR) Leicester Biomedical Research Centre (BRC). The authors also thank the Roy J Carver Charitable Trust, Muscatine, Iowa for ongoing research support.

## Additional information

### Funding

| Funder | Grant reference number | Author |
| --- | --- | --- |
| National Institute of Allergy and Infectious Diseases | AI43957 | Thomas E Wales John R Engen Amy H Andreotti |
| Roy J. Carver Charitable Trust | | Amy H Andreotti |
| Experimental Cancer Medicine Centre | C10604/A25151 | Sandrine Jayne Robert G Britton Martin JS Dyer |

The funders had no role in study design, data collection and interpretation, or the decision to submit the work for publication.

### Author contributions

Raji E Joseph, Conceptualization, Data curation, Formal analysis, Investigation, Validation, Visualization, Writing – original draft, Writing – review and editing; Thomas E Wales, Data curation, Formal analysis, Investigation, Writing – original draft, Writing – review and editing; Sandrine Jayne, Robert G Britton, Data curation, Formal analysis, Investigation, Writing – review and editing; D Bruce Fulton, Data curation, Formal analysis, Supervision, Writing – review and editing; John R Engen, Martin JS Dyer, Investigation, Funding acquisition, Conceptualization, Project administration; Amy H Andreotti, Formal analysis, Funding acquisition, Investigation, Project administration, Supervision, Writing – original draft, Writing – review and editing

### Author ORCIDs

Raji E Joseph ⓘ https://orcid.org/0009-0000-7122-8553
Thomas E Wales ⓘ https://orcid.org/0000-0001-6133-5689
Sandrine Jayne ⓘ https://orcid.org/0000-0003-1870-9782
John R Engen ⓘ https://orcid.org/0000-0002-6918-9476
Martin JS Dyer ⓘ https://orcid.org/0000-0002-5033-2236
Amy H Andreotti ⓘ https://orcid.org/0000-0002-6952-7244

Reviewer #1 (Public Review): https://doi.org/10.7554/eLife.95488.3.sa1
Reviewer #2 (Public Review): https://doi.org/10.7554/eLife.95488.3.sa2
Author response https://doi.org/10.7554/eLife.95488.3.sa3

## Additional files

### Supplementary files

Supplementary file 1. Bruton's tyrosine kinase (BTK) mutations detected in chronic lymphocytic leukemia (CLL) patients treated with BTK inhibitors. In addition to mutations in BTK C481, T474, and L528, other BTK mutations that have been detected include: R28S, E108K, G164D, V416L, A428D, M437R, R490H, Q516K, V537I, and T316A (*Maddocks et al., 2015*; *Ahn et al., 2017*; *Woyach et al., 2014*; *Blombery et al., 2022*; *Wang et al., 2022*; *Jackson et al., 2023*; *Handunnetti et al., 2019*; *Woyach et al., 2019*; *Woyach et al., 2017*; *Kanagal-Shamanna et al., 2019*; *Gángó et al., 2020*; *Sharma et al., 2016*).

Supplementary file 2. Excel file providing enhanced experimental details for hydrogen deuterium exchange mass spectrometry (HDX-MS) including minimum criteria specified by *Masson et al., 2019*, lists of all peptides by residue number, sequence, as well as deuterium levels measured for each Figure. The value of each deuterium difference for every colored box in each Figure as well as the complete dataset for each state are also found in this file.

MDAR checklist

### Data availability

All data generated or analyzed during this study are included in the manuscript and supporting files. All HDX-MS data have been deposited to the ProteomeXchange Consortium via the PRIDE partner repository with the dataset identifier PXD047865.

The following dataset was generated:

| Author(s) | Year | Dataset title | Dataset URL | Database and Identifier |
|---|---|---|---|---|
| Wales TE, Engen JR | 2024 | Impact of the clinically approved BTK inhibitors on the conformation of full-length BTK and analysis of the development of BTK resistance mutations in chronic lymphocytic leukemia | https://proteomecentral.proteomexchange.org/cgi/GetDataset?ID=PXD047865 | ProteomeXchange, PXD047865 |

The following previously published dataset was used:

| Author(s) | Year | Dataset title | Dataset URL | Database and Identifier |
|---|---|---|---|---|
| Wales TE, Engen JR | 2020 | Differential impact of BTK active site inhibitors on the conformational state of full-length BTK | https://proteomecentral.proteomexchange.org/cgi/GetDataset?ID=PXD020029 | ProteomeXchange, PXD020029 |

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
