## [Editor Report · eLife Assessment]

The manuscript reports on an **important** comparison of a range of approved clinical inhibitors for BTK used for the treatment of chronic lymphocytic leukemia (CLL). The authors provide **compelling** evidence for their claims, using a combination of HDX-MS and NMR spectroscopy. The novelty is that this study also seeks to evaluate resistance mutation bias. The manuscript will be of high interest to scientists working on protein drug interactions.

---

## [Referee Report · Reviewer #1 (Public Review)]

Summary:

The work by Joseph et al "Impact of the clinically approved BTK inhibitors on the conformation of full-length BTK and analysis of the development of BTK resistance mutations in chronic lymphocytic leukemia" seeks to comparatively analyze the effect of a range of covalent and noncovalent clinical BTK inhibitors upon BTK conformation. The novel aspect of this manuscript is that it seeks to evaluate the differential resistance mutations that arise distinctly from each of the inhibitors.

Strengths:

This is an exciting study that builds upon the fundamental notion of ensemble behavior in solutions for enzymes such as BTK. The HDX-MS and NMR experiments are adequately and comprehensively presented.

Comments on the revised version:

I am satisfied with the revisions and the clear explanations.

---

## [Referee Report · Reviewer #2 (Public Review)]

Summary:

Previous NMR and HDX-MS studies on full-length (FL) BTK showed that the covalent BTKi, ibrutinib, causes long-range effects on the conformation of BTK consistent with disruption of the autoinhibited conformation, based on HDX deuterium uptake patterns and NMR chemical shift perturbations. This study extends the analyses to four new covalent BTKi, acalabrutinib, zanubrutinib, tirabrutinib/ONO4059, and a noncovalent ATP competitive BTKi, pirtobrutinib/LOXO405.

The results show distinct conformational changes that occur upon binding each BTKi. The findings show consistent NMR and HDX changes with covalent inhibitors, which move helix aC to an 'out' position and disrupt SH3-kinase interactions, in agreement with X-ray structures of the BTKi complexed with the BTK kinase domain. In contrast, the solution measurements show that pirtobrutinib maintains and even stabilizes the helix aC-in and autoinhibited conformation, even though the BTK:pritobrutinib crystallizes with helix aC-out. This and unexpected variations in NMR and HDX behavior between inhibitors highlight the need for solution measurements to understand drug interactions with the full-length BTK. Overall the findings present good evidence for allosteric effects by each BTKi that induce distal conformational changes which are sensitive to differences in inhibitor structure.

The study goes on to examine BTK mutants T474I and L528W, which are known to confer resistance to pirtobrutinib, zanubritinib, and tirabrutinib. T474I reduces and L528W eliminates BTK autophosphorylation at pY551, while both FL-BTK-WT and FL-BTK-L528W increase HCK autophosphorylation and PLCg phosphorylation. These show that mutants partially or completely inactivate BTK and that inactive FL-BTK can activate HCK, potentially by direct BTK-HCK interactions. But they do not explain drug resistance. However, HDX and NMR show that each mutant alters the effects of BTKi binding compared to WT. In particular, T474I alters the effects of all three inhibitors around W395 and the activation loop, while L528W alters interactions around W395 with tirabrutinib and pirtobrutinib, and does not appear to bind zanubrutinib at all. The study concludes that the mutations might block drug efficacy by reducing affinity or altering binding mode.

Strengths:

The work presents convincing evidence that BTK inhibitors alter the conformation of regions distal to their binding sites, including those involved in the SH3-kinase interface, the activation loop, and a substrate binding surface between helix aF and helix aG. The findings add to the growing understanding of allosteric effects of kinase inhibitors, and their potential regulation of interactions between kinase and binding proteins.

Comments on the revised version:

The authors have satisfactorily addressed my concerns in their revised manuscript.

---

## [Author Response]

The following is the authors’ response to the original reviews.

**Public Reviews:**

**Reviewer #1 (Public Review):**
Summary:The work by Joseph et al "Impact of the clinically approved BTK inhibitors on the conformation of full-length BTK and analysis of the development of BTK resistance mutations in chronic lymphocytic leukemia" seeks to comparatively analyze the effect of a range of covalent and noncovalent clinical BTK inhibitors upon BTK conformation. The novel aspect of this manuscript is that it seeks to evaluate the differential resistance mutations that arise distinctly from each of the inhibitors.Strengths:This is an exciting study that builds upon the fundamental notion of ensemble behavior in solutions for enzymes such as BTK. The HDX-MS and NMR experiments are adequately and comprehensively presented.

We thank the reviewer for this positive feedback.

Weaknesses:While I commend the novelty of the study, the absence of important controls greatly tempers my enthusiasm for this work. As stated in the abstract, there are no broad takeaways for how resistance mutation bias operated from this study, although the mechanism of action of 2 common resistance mutations is useful. How these 2 resistance mutations connect to ensemble behavior, is not obvious. This is partly because BTK does not populate just binary "open"/"closed" conformations, but there are likely multiple intermediate conformations. Each inhibitor appears to preferentially "select" conformations by the authors' own assessment (line 236) and this carries implications for the emergence of resistance mutations. The most important control that would help is to use ADP or nonhydrolyzable and ATP as a baseline to establish the "inactive" and "active" conformations. All of the HDX-MS and NMR studies use protein that has no nucleotide present. A major question that remains is whether each of the inhibitors preferentially favors/blocks ADP or ATP binding. This then means it is not equivalent to correlate functional kinase assay conditions with either HDX-MS or NMR experiments.

We thank the reviewer for raising this point. The BTK inhibitors studied here are active site inhibitors that completely prevent (block) nucleotide (both ATP and ADP) binding. We believe the other question being asked here is whether the different BTK inhibitors bind preferentially to the ADP or ATP bound kinase (do the conformational states favored by ADP versus ATP bound BTK affect drug binding). We agree this is an interesting question that deserves further study. Here we are focused on the ligand bound state itself rather than on the conformational state selection mechanism of each inhibitor. Thus, HDX-MS and NMR work to compare ligand bound to apo-, ADP, and ATP bound BTK is beyond the scope of this manuscript. That said, previous work (doi: 10.1038/s41598-017-17703-5) has shown that the related TEC kinase, ITK, preferentially binds ADP when the kinase is in the autoinhibited conformation. Since we have previously shown that BTK adopts the autoinhibited conformation in the nucleotide free form (https://doi.org/10.7554/eLife.89489.2), we suggest that the comparison we have carried out here between drug bound and apo-protein is valid. Future work will carefully address the conformational preferences of all three conditions, apo-, ADP- and ATP-bound.

**Reviewer #2 (Public Review):**
Summary:Previous NMR and HDX-MS studies on full-length (FL) BTK showed that the covalent BTKi, ibrutinib, causes long-range effects on the conformation of BTK consistent with disruption of the autoinhibited conformation, based on HDX deuterium uptake patterns and NMR chemical shift perturbations. This study extends the analyses to four new covalent BTKi, acalabrutinib, zanubrutinib, tirabrutinib/ONO4059, and a noncovalent ATP competitive BTKi, pirtobrutinib/LOXO405.The results show distinct conformational changes that occur upon binding each BTKi. The findings show consistent NMR and HDX changes with covalent inhibitors, which move helix aC to an 'out' position and disrupt SH3-kinase interactions, in agreement with X-ray structures of the BTKi complexed with the BTK kinase domain. In contrast, the solution measurements show that pirtobrutinib maintains and even stabilizes the helix aC-in and autoinhibited conformation, even though the BTK:pritobrutinib crystallizes with helix aC-out. This and unexpected variations in NMR and HDX behavior between inhibitors highlight the need for solution measurements to understand drug interactions with the full-length BTK. Overall the findings present good evidence for allosteric effects by each BTKi that induce distal conformational changes which are sensitive to differences in inhibitor structure.The study goes on to examine BTK mutants T474I and L528W, which are known to confer resistance to pirtobrutinib, zanubritinib, and tirabrutinib. T474I reduces and L528W eliminates BTK autophosphorylation at pY551, while both FL-BTK-WT and FL-BTK-L528W increase HCK autophosphorylation and PLCg phosphorylation. These show that mutants partially or completely inactivate BTK and that inactive FL-BTK can activate HCK, potentially by direct BTK-HCK interactions. But they do not explain drug resistance. However, HDX and NMR show that each mutant alters the effects of BTKi binding compared to WT. In particular, T474I alters the effects of all three inhibitors around W395 and the activation loop, while L528W alters interactions around W395 with tirabrutinib and pirtobrutinib, and does not appear to bind zanubrutinib at all. The study concludes that the mutations might block drug efficacy by reducing affinity or altering binding mode.Strengths:The work presents convincing evidence that BTK inhibitors alter the conformation of regions distal to their binding sites, including those involved in the SH3-kinase interface, the activation loop, and a substrate binding surface between helix aF and helix aG. The findings add to the growing understanding of allosteric effects of kinase inhibitors, and their potential regulation of interactions between kinase and binding proteins.

We thank the reviewer for these positive comments.

Weaknesses:The interpretation of HDX, NMR, and kinase assays is confusing in some places, due to ambiguity in quantifying how much kinase is bound to the inhibitor. It would be helpful to confirm binding occupancy, in order to clarify if mutants lower the amount of BTK complexed with BTKi as implied in certain places, or if they instead alter the binding mode. In addition, the interpretation of the mutant effects might benefit from a more detailed examination of how each inhibitor occupies the ATP pocket and how substitutions of T474 and L528 with Ile and Trp respectively might change the contacts with each inhibitor.

We thank the reviewer for these suggestions. As requested we have now modified the manuscript to clearly state the effects of the mutations on inhibitor binding. Additionally, we have included a new figure to discuss the interaction of the inhibitors within the BTK kinase active site to provide a better explanation for the impact of the resistance mutations.

**Recommendations for the authors:**

**Reviewer #1 (Recommendations For The Authors):**
Major Comments:(1) What is the binding affinity of ATP/ADP to BTK? BTK is purified by the authors as an apoenzyme (by the final purification by SEC, all protein should be completely stripped of nucleotide)- but must toggle between ATP and ADP-bound states. Do the inhibitors completely sterically block nucleotide binding? Do they only block one or the other- ADP/ATP binding? Do they weaken ADP/ATP binding? The authors have an opportunity with NMR to establish a clear baseline to compare the inhibitors' effects on BTK. It is not clear if the authors' assumption is that all BTKi share a common mode of action (Line 114).

All BTK inhibitors studied in this work (Ibrutinib, Acalabrutinib, Zanubrutinib, Tirabrutinib and Pirtobrutinib) share a common mode of action. They are active site inhibitors that completely block nucleotide (ATP and ADP) binding. The introduction to the manuscript has been updated to add this information (lines 70-71, pg. 4).

"The covalent BTK inhibitors (Ibrutinib, Acalabrutinib, Zanubrutinib and Tirabrutinib) and the non-covalent BTK inhibitor Pirtobrutinib bind tightly to the BTK active site (Kinact/KI or KD values in the nM range; DOI: 10.1056/NEJMoa2114110). In contrast, previous studies have reported nucleotide affinity for TEC kinases that are lower (KD in the µM range), (doi: 10.1038/s41598-017-17703-5). Additionally, the same work has shown that the conformational state of TEC kinases can impact nucleotide binding. The TEC kinases have a higher affinity for ADP (KD ~ 20 µM), as compared to ATP (KD ~ 15 fold lower than ADP), when the full-length protein adopts the autoinhibited conformation. Disruption of the TEC kinase autoinhibited conformation (by mutation) decreases the affinity for ADP, allowing ATP to bind, enabling kinase activity. Nevertheless, regardless of the conformational state of BTK, all the BTK inhibitors studied here block both ADP and ATP binding to the active site."

(2) Is there an effect of nucleotide binding bias on resistance mutation emergence? Is there a nucleotide binding bias in the resistance mutations characterized in this study? There likely is - BTK L528W is catalytically inactive. It is not clear if this mutant stays bound to ADP or to ATP and cannot transfer the phosphate to its substrate. How does BTK T474I interact with ADP/ATP? This is needed before concluding - in lines 289-291- that mutations cause only minor conformational changes. This needs a qualifier - in the nucleotide-free apo conformation.

The BTK L528W mutation introduces a bulky sidechain into the BTK kinase active site that sterically impedes both ATP and ADP binding. In fact, previous studies (https://doi.org/10.1016/j.jbc.2022.102555) have confirmed the inability of the BTK L528W mutant to bind ATP.

The BTK T474I mutation could alter nucleotide binding. However, The BTK T474I mutation lowers the overall activity of BTK, and is consistent with previous work that have shown the same (https://doi.org/10.1021/acschembio.6b00480). The decrease in overall kinase activity cannot account for the development of resistance (which typically requires increased kinase activity). Hence, a decrease in inhibitor binding is likely driving resistance.

Lines 293 (pg. 14) have been modified to indicate that the conformational changes observed in the BTK mutants are in the absence of nucleotide as requested.

(3) What is the half-life BTK? And does inhibitor binding to BTK change the half-life of the inhibitor?

BTK has a long half-life of 48-72 h (DOI: https://doi.org/10.1124/jpet.113.203489). Unbound covalent inhibitors are rapidly cleared from the body with short half-lives on the order of < 4h. Non-covalent BTK inhibitors typically have a longer half-life on the order of 20h. Once bound to BTK, the irreversible nature of binding by covalent inhibitors make them unavailable to other molecules. CLL patients are treated typically with a once daily or twice daily dose of BTK inhibitor. Hence, inhibitor binding to BTK does not alter the half-life of free inhibitor.

(4) Are there broad differences between covalent and single non-covalent inhibitors upon resistance mutation bias? And nucleotide binding?

The biggest difference observed between BTK covalent and non-covalent inhibitors in the emergence of resistance mutations is the occurrence of the C481S mutation in patients treated with covalent inhibitors. This resistance mutation is absent in patients treated with non-covalent BTK inhibitors. Patients thatdevelop mutations in BTK C481 can no longer be treated with any of the approved covalent BTK inhibitors (as they all use BTK C481 for covalent linkage). To ensure BTK inhibition, patients with mutations in C481 can be treated with non-covalent BTK active site inhibitors. All currently approved BTK inhibitors (covalent and non-covalent) are active site inhibitors that compete with nucleotide binding.

(5) It's unclear why the authors chose to evaluate the impact of inhibitor binding on the linker kinase domain first. This seems unnecessary.

NMR analysis is easier on the smaller BTK linker kinase domain (LKD) fragment compared to the full-length protein. Hence for practical reasons we used the BTK LKD fragment.

(6) Line 508 - there seems to be a gap in understanding protein half-lives, inhibitor half-lives, and the emergence of resistance mutations in this manuscript itself. The manuscript falls short of a mechanistic descriptor of variable inhibitors and resistance mutation bias.

The half-life of the inhibitors assessed in this study are provided in Table 1 of this manuscript. The emergence of resistance mutations such as C481 are likely due to a direct consequence of differences in inhibitor half-life as described in the discussion section of this manuscript (page 23).

(7) HDX-MS reports the conformational average difference across the ensemble but does not distinguish between the number of intermediary conformations. The authors should clarify that this is a limitation of an average readout method such as HDX-MS. This is currently not addressed.

A sentence describing this limitation has been added (lines 219-221, pg. 11) as requested.

Minor Points:(1) Some of the qualitative descriptors are unnecessary - line 284 - "Slightly towards....". Line 286 - "Slight stabilizing effect on the conformation..." How slight is slight?

Qualitative descriptors have been removed from the manuscript as requested.

(2) The authors should provide SPR data with Kon and Koff values for Pirtobrutinib binding to BTK (in the presence of ARP and ADP).

SPR analysis of Pirtobrutinib has previously been reported. Pirtobrutininb binds to BTK wild-type with a KD of 0.9 nM (DOI: 10.1056/NEJMoa2114110). As mentioned earlier in response to comment 1, Pirtobrutinib binds to the BTK kinase active site and is competitive with both nucleotides (ATP and ADP, which bind with lower affinity, KD in the µM range).

(3) In Figure 2, the legend needs to describe the specific time point represented. Same with Figure 5.

The HDX-MS changes that are mapped onto the structure represent the maximal changes observed at any time point. The figure legends have been modified as requested to clarify this.

**Reviewer #2 (Recommendations For The Authors):**
(1) Figure 7 is an amazing and impressive finding, but it could use two controls: First a blot of pY551 to show more rigorously that FL-BTK-WT and L528W autophosphorylation is unaffected by zanubrutinib binding, just to eliminate the possibility that elevated pY551 accounts for the enhanced HCK phosphorylation.

Both BTK FL enzymes (WT and L528W) in this assay are catalytically inactive and do not contribute to autophosphorylation on BTK Y551 (BTK FL WT is inhibited by Zanubrutinib and BTK FL L528W is catalytically dead). Additionally, BTK FL WT and BTK FL L528W are both able to activate HCK. Hence differences in pY551 levels between these BTK proteins cannot explain how both proteins are able to activate HCK.

Nevertheless, as requested, we probed for pY551 levels on BTK. While BTK cannot autophosphorylate itself on BTK Y551 in this assay, BTK Y551 is able to be phosphorylated by HCK. BTK Y551 phosphorylation levels were higher in BTK FL WT compared to BTK FL L528W likely due to Y551 on the activation loop being less accessible in the BTK L528W mutant (which is more stabilized in the autoinhibited conformation) compared to the WT protein. This data has been added as a new panel in Figure 7a.

Additionally, we tested the ability of the BTK FL L528W/Y551F double mutant to activate HCK. The BTK FL L528W/Y551F double mutant is able to activate HCK similar to BTK FL L528W single mutant, demonstrating that phosphorylation on Y551 is not necessary for HCK activation by BTK FL L528W. This new data has been added as supplemental figure S2a. Taken together, pY551 levels on BTK do not contribute to enhanced HCK phosphorylation. The results section of the manuscript has been modified to include this additional data (Lines 319-335, pg. 15-16).

Second, controls performed in the absence of Zanubrutinib are needed for the time courses with HCK alone, HCK + FL-BTK WT, and HCK + FL-BTK-L528W. This would help show that the ability of BTK to increase the phosphorylation of HCK and PLCg1 is (or isn't) dependent on drug interactions with BTK, HCK, or PLCg.

BTK FL L528W can enhance phosphorylation on PLCg by HCK even in the absence of Zanubrutinib. We have added this data as a new supplemental figure S2b. We have not included BTK FL WT in this analysis as in the absence of Zanubrutinib, we would have two active enzymes (HCK and BTK) in the assay which would complicate the interpretation of the data. The results section of the manuscript has been modified to include this additional data (Lines 333-335, pg. 16).

And please comment: in cells, does zanubrutinib treatment (or any other drug) increase pY phosphorylation of HCK or PLCg?

All clinically approved BTK inhibitors (covalent and non-covalent) inhibit BTK WT activity and decrease PLCg phosphorylation in cells. There have been no reports, to our knowledge, of any clinically approved BTK inhibitor causing an increase in HCK activity.

(2) Sections of the Results discussing Figures 8 and 9 are confusing to read because they variously propose that the mutants (i) reduce inhibitor occupancy, or (ii) alter the inhibitor binding mode. However, some of the results unambiguously show an altered binding mode instead of reduced inhibitor binding.a) For example, HDX clearly shows protection by tira, zanu, and pirto, therefore reduced inhibitor binding does not seem to be an option. Therefore, I recommend modifying lines 357-363. "The differences in deuterium exchange for drug binding to WT and mutant BTK suggest that the T474I mutation either causes a reduction in inhibitor binding or otherwise alters the mode of drug interaction in the active site. "

While the HDX-MS data of BTK T474I shows protection by Tirabrutinib, Zanubrutinib and Pirtobrutinib, the magnitude of the protection is reduced in the BTK T474I mutant compared to WT BTK (Fig. 8e) suggesting a reduction in inhibitor binding. These results are consistent with previous SPR analysis of the BTK T474I mutant which also showed reduced binding to Zanubrutinib, Acalabrutinib and Pirtobrutinib (DOI: 10.1056/NEJMoa2114110). The manuscript (lines 381-383, pg. 18) has been modified to clearly state that the BTK T474I mutation causes a reduction in inhibitor binding.

b) I recommend modifying lines 370-373." In stark contrast to the BTK T474I mutant, the BTK 370 L528W mutant does not show any change in deuterium incorporation in the presence of 371 Zanubrutinib, Tirabrutinib or Pirtobrutinib, providing strong evidence that the BTK L528W 372 mutant does not bind the inhibitors (Fig.8d)."Lines 432-435: Although the L528W mutation alters binding to both Tirabrutinib 432 and Pirtobrutinib, the NMR data suggests that it retains partial binding unlike the HDX-MS data 433 that suggests complete disruption of binding. The higher inhibitor concentrations used in the NMR 434 experiments compared to the HDX-MS experiments likely explain this discrepancy."The discordance in the L528W mutant between the lack of any HDX protection by tira and pirto versus the clear chemical shift of W395 by NMR is worrisome. If the HDX experiments were really done under conditions where binding occupancy was too low, then it seems important to redo these experiments at higher drug concentrations.Alternatively, and perhaps more useful would be to report Kd for binding of these inhibitors to the two mutants. That would allow the authors to interpret these results more definitively.

SPR analysis of inhibitor binding to full-length BTK WT, T474I and L528W has been previously reported (DOI: 10.1056/NEJMoa2114110). The covalent BTK inhibitors (Ibrutinib, Acalabrutinib, and Zanubrutinib) and the non-covalent BTK inhibitor Pirtobrutinib bind tightly to full-length WT BTK (Kinact/KI or KD values in the nM range). The BTK T474I mutation disrupts binding to Zanubrutinib, Acalabrutinib and Pirtobrutinib, but not Ibrutinib and Fenebrutinib. BTK L528W mutation disrupts binding to Zanubrutinib, Acalabrutinib, Ibrutinib and Pirtobrutinib, but not Fenebrutinib. These previously published results are consistent with the HDX-MS and NMR data presented here. The manuscript has been modified to clearly state that the mutations reduce drug binding instead of altered binding.

c) Recommend adding data to confirm statements in lines 419-421:"Spectral overlays of the BTK L528W mutant with and without Zanubrutinib show no 419 chemical shift changes (Fig. 9a, right panel) suggesting that the mutation completely disrupts 420 inhibitor binding in complete agreement with the HDX-MS data (Fig. 8d).428-432: The Pirtobrutinib-bound BTK L528W spectrum (Fig. 9c) shows two resonance positions, 428 one of which overlaps with the W395 resonance in the apo protein and the other that corresponds to that of the mutant protein bound to Pirtobrutinib. This data suggests a mixture of inhibitor bound and unbound BTK kinase domain in solution, likely due to a reduction in Pirtobrutinib affinity 431 caused by the L528W mutation."Likewise, direct measurements of binding affinity to L528W would be helpful. It is not completely convincing that the effects of this mutant are due to the reduced binding of either inhibitor. The effects of pirtobrutinib may instead reflect a slow exchange of W395 instead of 50% occupancy. For example, what happened in the rest of the spectra? Were other chemical shifts apparent in either case, which might address binding stoichiometry? It would be useful to show the full spectra in Supplemental figures, as well as any titrations that may have been done to confirm that the inhibitors are added at saturating concentration.

As requested the full-spectra of Pirtobrutinib bound to BTK L528W has now been added as supplemental figure S1c. In the BTK L528W bound to Pirtobrutinib spectrum, two cross peaks are visible for multiple resonances, one of which overlaps with that of the apo BTK L528W spectrum, suggesting that there is a mixture of apo and inhibitor bound forms of BTK L528W.

The clinically approved inhibitors that we are working with here (Ibrutinib, Acalabrutinib, Zanubrutinib, Tirabrutinib and Pirtobrutinib have reported IC50 values in the nM range 0.5 nM, 3 nM, 0.3 nM, 6.8 nM and 3.68 nM respectively). All the NMR work presented here was carried out at a 1:1.33, protein:inhibitor ratio (absolute concentration of the inhibitor was 200 µM). NMR titrations of BTK WT have been carried out with Ibrutinib (https://doi.org/10.7554/eLife.60470) and Tirabrutinib. Complete binding is observed at a 1:1 molar ratio of protein:inhibitor, consistent with the previously reported binding characteristics. Mass spec analysis also shows one covalent inhibitor bound to each BTK WT protein (Fig. 4a). The BTK T474I and L528W mutants were tested at the same protein:inhibitor ratio as WT BTK for ease of comparison.

(3) The Discussion could use a structural perspective on the likely effects of each mutation on inhibitor binding. Both residues occupy positions in beta7 and the hinge, which are commonly found to form hydrophobic and polar contacts with ATP competitive inhibitors in many kinases. This would be useful to discuss and show as a figure, in order to give the non-kinase expert a better understanding of why the mutations might affect inhibitor binding. The variations in structures of each inhibitor and how they contact these two positions might be useful to inspect, and ask why some inhibitors but not others are affected by mutation, and why some inhibitors but not others induce effects over long distances to W395 and the activation loop.

As requested, we have added a new paragraph in the discussion and a new figure (Fig. 10), to expand on likely effects of the mutations on inhibitor binding. The allosteric effects of some of the BTK inhibitors, on the other hand are currently being investigated and is beyond the scope of the current manuscript.

(4) The authors propose that small differences in Tm and stability of L358W account for its effect on resistance. Does this mutant show elevated expression in patient tumors over those with WT BTK?

Preliminary data indicates that BTK L528W levels are elevated in one of two patients carrying this resistance mutation. However, due to the low number of patients tested, we have chosen to not include the data in this study but will continue to pursue this question in future work.